# Reflection-Bench: Evaluating Epistemic Agency in Large Language Models

**Lingyu Li** [1 2]  **Yixu Wang** [1]  **Haiquan Zhao** [1]  **Shuqi Kong** [1 2]  **Yan Teng** [1]  **Chunbo Li** [2]  **Yingchun Wang** [1]

## Abstract

With large language models (LLMs) increasingly deployed as cognitive engines for AI agents, the reliability and effectiveness critically hinge on their intrinsic epistemic agency, which remains understudied. Epistemic agency, the ability to flexibly construct, adapt, and monitor beliefs about dynamic environments, represents a base-model-level capacity independent of specific tools, modules, or applications. We characterize the holistic process underlying epistemic agency, which unfolds in seven interrelated dimensions: prediction, decision-making, perception, memory, counterfactual thinking, belief updating, and meta-reflection. Correspondingly, we propose Reflection-Bench, a cognitive-psychology-inspired benchmark consisting of seven tasks with long-term relevance and minimization of data leakage. Through a comprehensive evaluation of 16 models using three prompting strategies, we identify a clear three-tier performance hierarchy and significant limitations of current LLMs, particularly in meta-reflection capabilities. While state-of-the-art LLMs demonstrate rudimentary signs of epistemic agency, our findings suggest several promising research directions, including enhancing core cognitive functions, improving cross-functional coordination, and developing adaptive processing mechanisms. Our code and data are available at https://github.com/AI45Lab/ReflectionBench.

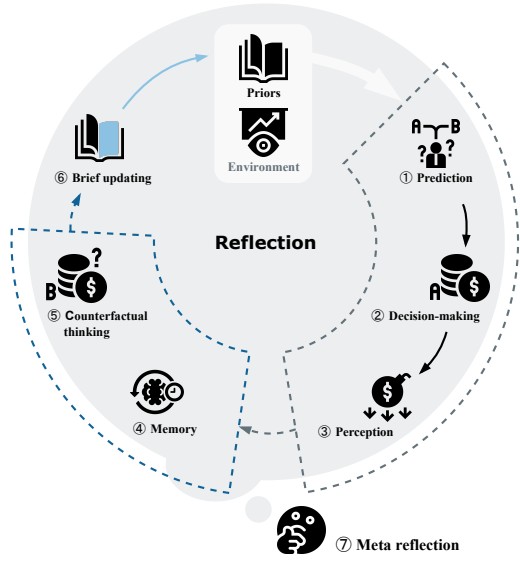

Figure 1. Reflection, a cyclic cognitive process underlying epistemic agency, enables effective agent-environment interaction.

## 1. Introduction

Large language models (LLMs) have been increasingly applied as the brains for AI agents, interacting with the environment and completing various tasks such as programming, scientific research, and industrial production. (Xi et al., 2023; Wang et al., 2024; Schmidgall et al., 2025). The performance of these LLMs-based agents fundamentally depends on high-order capacities of models, such as reasoning, planning, recovering from errors, and learning from the environment, etc.(Schluntz & Zhang, 2024). Collectively, these abilities reflect a model's intrinsic *epistemic agency* – the capacity to flexibly construct, adapt, and monitor beliefs about dynamic environments (Ferrero, 2022; Fairweather & Montemayor, 2017; Chrisman, 2022). This epistemic agency, as a base-model-level characteristic independent of specific external modules or applications, determines whether LLMs can truly serve as reliable cores of AI agents. However, existing studies either focus narrowly on agents' applications or examine isolated abilities (Liu et al., 2023; Clark et al., 2018; Valmeekam et al., 2024). None of them investigates LLMs' epistemic agency unfolding in the holistic process of agent-environment interaction. Furthermore, current evaluations' reliance on text-heavy datasets introduces methodological limitations, particularly the risks of

---

Work done during Lingyu's internship at Shanghai Artificial Intelligence Laboratory. [1]Shanghai Artificial Intelligence Laboratory, China [2]Shanghai Mental Health Center, Shanghai Jiao Tong University School of Medicine, China. Correspondence to: Yan Teng <tengyan@pjlab.org.cn>, Chunbo Li <licb@smhc.org.cn>.

benchmark leakage (Zhou et al., 2023; Xu et al., 2024).

To evaluate LLMs' epistemic agency, we propose *Reflection-Bench*, a cognitive-psychology-inspired benchmark with minimization of data contamination. Our framework is motivated by mental processes of how human epistemic agency enables efficient agent-environment interaction. As illustrated in Figure 1, this cyclical process involves predicting based on prior belief, making decisions to achieve desired states, perceiving surprises, subsequently modifying priors through recalling and counterfactual thinking (Friston et al., 2017; Parr et al., 2022; Dennett, 1993; Friston, 2010; Huang & Rao, 2011). The complete cycle embodies *reflection* – the "attentive, critical, exploratory and iterative interactions with one's thoughts and actions with the intention to change them (Nguyen et al., 2014)." We therefore adopt reflection, defined here as a process rather than an ability, as our framework to investigate LLMs' epistemic agency. Through detailed analysis of this reflection process, we identify seven key dimensions that constitute robust epistemic agency, including prediction, decision-making, perception, memory, counterfactual thinking, belief updating, and meta-reflection.

To evaluate these seven dimensions, we design the Reflection-Bench with seven *cognitive tests* adapted from validated paradigms in cognitive psychology. Cognitive tests create controlled environments where subjects must learn and reason about unknown parameters through interaction, offering standardized, quantified, and objective assessment tools that mirror real-world functioning (Lezak et al., 2012; Allen et al., 2024). We select and adapt seven cognitive tests for LLMs evaluation, each focusing on one function shown in Figure 1 (Nyhus & Barceló, 2009; Shohamy et al., 2008; Näätänen et al., 2007; Jaeggi et al., 2010; Buelow & Suhr, 2009; Bouneffouf et al., 2020; Bechara et al., 2013; Cools et al., 2002). The parameterized design of cognitive tests allows their configurations to be customized – even if a model has memorized the task format, it cannot know the specific parameters used in evaluation. This design both reduces potential data contamination and ensures Reflection-Bench's long-term relevance by enabling difficulty adjustment to accommodate future AI models.

We conducted comprehensive evaluations using entry-level configurations across leading large reasoning models, mainstream LLMs, and the Qwen-2.5 family with varying sizes. Three prompting strategies were employed, including direct generation, free output, and Chain of Thought (CoT) (Wei et al., 2024). The results reveal a clear three-tier hierarchy, with seven state-of-the-art models scoring over 60, eight moderate models between 50 and 60, and the model with the smallest size scoring below 50 points. While these results demonstrate certain epistemic agency in current LLMs, detailed behavioral analyses reveal significant

limitations of LLMs' epistemic agency, especially in prediction, decision-making, and meta-reflection. The effectiveness of prompting strategies varies substantially across both tasks and models. We further evaluated these models on a more challenging parameter set using the direct generation strategy. The expected score decrease suggests that Reflection-Bench is far from saturated. Additionally, we compared the performances of Centaur, which is specifically fine-tuned with human cognitive test data (Binz et al., 2024), and its base model, Llama-3.1-70B-Instruct. Centaur shows no improvement compared to its base model, providing empirical evidence that our parameterized design effectively minimizes data leakage concerns.

To summarize, our main contributions include:

- Development of Reflection-Bench, a cognitive-psychology-inspired benchmark that evaluates LLMs' epistemic agency through parameterized and contamination-minimization tasks.

- Comprehensive evaluations across 16 LLMs using three prompting strategies, revealing distinct performance tiers and demonstrating Reflection-Bench's effectiveness in differentiating models' epistemic agency.

- Multi-dimensional behavioral analyses across models, tasks, and prompting strategies, uncovering both emerging epistemic agency and fundamental limitations in current LLMs, and providing insights for future research directions.

## 2. Related work

### 2.1. LLMs-based Agents

The concept of *Agent* originally describes several key aspects of biological systems: (1) flexibly interacting with the environment to achieve a range of goals; (2) making choices actively; and (3) being accountable for the outcomes of their actions (Kockelman, 2006), and higher *agency* manifests as enhanced sophistication of these three aspects. With the emergence of high-order capabilities such as understanding and reasoning (Wei et al., 2022), LLMs are increasingly deployed as agents, where they "dynamically direct their own process and tool usage, maintaining control over how they accomplish tasks" (Schluntz & Zhang, 2024). Similar to biological agents, these LLMs-based agents work in interactive action-feedback loops to complete complex tasks, operating on various architectures for better performances than naïve LLMs (Yao et al., 2022; Nakajima, 2023; Hong et al., 2023). Correspondingly, multiple benchmarks have been developed for evaluating these agents' capabilities of performing specific tasks such as shopping, gaming, UI interface, customer service, travel, jailbreak-proof, and so on

(Liu et al., 2023; Deng et al., 2024; Xiao et al., 2024; Andriushchenko et al., 2024). Apart from application-specific evaluations, prior research has examined fundamental capabilities required for LLMs to function as agents, including reasoning (Clark et al., 2018), planning (Valmeekam et al., 2024), causal inference (Chen et al., 2024a), probability estimation (Krishnamurthy et al., 2024), cognitive flexibility (Wilie et al., 2024), decision making (Li et al., 2024a), and spatial cognition (Madge & Poesio, 2024) etc. In this work, we specifically focus on epistemic agency, the ability to flexibly construct, adapt, and monitor beliefs about dynamic environments (Ferrero, 2022; Fairweather & Montemayor, 2017; Chrisman, 2022). This base-model-level capacity, independent of specific external modules or applications, directly determines whether a model can reliably serve as the "brain" of an AI agent.

## 2.2. Cognitive Psychology Methods for Evaluating LLMs

Cognitive psychology, which aims to understand human behavioral and mental processes (Sternberg & Sternberg, 2006), provides an effective framework for interpreting LLMs' capabilities. As LLMs grow in complexity, researchers increasingly adapt cognitive psychology methods to evaluate them. The core idea behind it is to treat LLMs as humanoid subjects in psychology experiments to probe their capabilities and mechanisms of cognitive traits (Binz & Schulz, 2023). Diverse paradigms and tests in cognitive psychology have been adapted for evaluating LLMs, including psychometrics for attributes (Li et al., 2024b), iterated learning for prior knowledge (Zhu & Griffiths, 2024), Wisconsin card sorting test and letter number test for cognitive flexibility (Kennedy & Nowak, 2024), implicit statistical learning and facial recognition etc. for effects of CoT (Liu et al., 2024b), Montreal Cognitive Assessment for cognitive impairment (Dayan et al., 2024), Cognitive Reflection Test for human-like intuition (Hagendorff et al., 2023) and so on. This method provides a top-down perspective toward explainable AI through behavioral analysis rather than direct interpretation of neural network activities (Hagendorff, 2023). Furthermore, just as we trust humans in everyday tasks based on their demonstrated cognitive capabilities and behavioral patterns, this approach potentially helps establish similar trust mechanisms for LLMs-based agents (Shiffrin & Mitchell, 2023).

## 3. Reflection-Bench

### 3.1. Define Dimensions of Epistemic Agency

To systematically evaluate epistemic agency, we identify key dimensions by analyzing the cognitive process of how agents implement robust interaction with dynamic environments. Through the lens of cognitive psychology, we decompose the epistemic agency into seven core capabilities that emerge in temporal order, as illustrated in Figure 1.

The process begins with agents' prior beliefs about the environmental states manifesting in LLMs as both world model (Yildirim & Paul, 2024) and task-specific expectations through zero/few-shot comprehension (Brown et al., 2020). Priors help establish a framework of task objectives, constraints, possible inputs and outputs, etc. **Prediction**, computed via transition probability (Friston, 2010) enables agents to hypothesize future states given potential actions. For LLMs, prediction capabilities are crucial for planning tasks, where models must reason about which policies will effectively transition an agent from its initial state to a desired goal state. (Valmeekam et al., 2024; Fu et al., 2024). Then, agents will determine concrete actions to execute based on prediction, i.e., **decision-making** (Shadlen & Kiani, 2013), constituting the material foundation for agent-environment interaction.

After policy execution, agents **perceive** environmental feedback, where the most valuable information is discrepancies from predictions, such as errors and penalties – a basic feature underlying both human cognition and reinforcement learning (Den Ouden et al., 2012). As crucial signals driving modification and adaptation, the prediction errors potentially originate from deviant internal models or changed environments. Two mental processes are triggered by prediction errors: recalling the past process and simulating choices not taken, (Byrne, 2016), i.e., **memory** and **counterfactual thinking**, respectively.

Through these two retrospective analyses, agents **update** their prior **beliefs** that more accurately describe the environment to perform better in the future (Parr et al., 2022). In LLMs, this error-driven learning mechanism manifests through in-context learning and causal reasoning, enabling dynamic belief adjustment to recover from errors (Dong et al., 2024; Chen et al., 2024a; Krishnamurthy et al., 2024; Schluntz & Zhang, 2024). Beyond such instant interactions, agents like humans also possess higher-order capabilities of meta-cognition – monitor, evaluate, and regulate their own cognitive process (Martinez, 2006). When it comes to the reflection process defined above, the **meta-reflection** enables agents to transcend local adaptation by analyzing patterns across multiple prediction-action-feedback cycles, thereby grasping global patterns of the environment.

To summarize, we identify seven interlinked cognitive dimensions that constitute robust epistemic agency: prediction, decision-making, perception, memory, counterfactual thinking, belief updating, and meta-reflection. This process-oriented framework enables systematic evaluation of LLMs' epistemic agency through well-defined cognitive tests.

## 3.2. Select Cognitive Tests

While these seven dimensions are interconnected in agent-environment interaction, cognitive psychology provides validated test paradigms that isolate and measure specific cognitive capabilities while controlling other dimensions (Lezak et al., 2012). This enables our evaluation of each dimension through its corresponding cognitive test, as illustrated in Figure 2.

**Prediction**  We select the *weather prediction task* (WPT) paradigm, where subjects originally predict the weather based on card cues by learning probabilistic relationships (Shohamy et al., 2008). We adapt this paradigm by converting these relationships into explicitly defined transition probability matrices. Models must learn how the cues influence the weather transitions, allowing us to assess their prediction capability.

**Decision-making**  We employ the *Wisconsin card sorting test* (WCST) to assess flexible decision-making (Nyhus & Barceló, 2009). Participants match cards varying in color, shape, and number of figures according to an undisclosed rule (e.g., by color) that must be inferred from feedback. The matching rule changes without warning after a set number of trials, requiring models to infer and adapt to the latent rule governing correct decisions.

**Perception**  We select the *oddball paradigm* (Näätänen et al., 2007) to evaluate the automatic perception of surprise signals. The paradigm presents sequences containing frequent standards interspersed with rare deviant stimuli (*e.g.*, different tones), measuring the automatic detection of novelty. Research shows that the brain is naturally sensitive to deviant stimuli (Garrido et al., 2009), making this paradigm ideal for assessing basic surprise detection. Impaired performance on this test is a reliable indicator of cognitive deficits in conditions like schizophrenia (Umbricht & Krljes, 2005).

**Memory**  We adopt the *n-back* task to evaluate active memory retrieval (Jaeggi et al., 2010). In this task, subjects view sequential stimuli and must indicate whether the current stimulus matches the one presented n steps earlier. This continuous process of updating and maintaining information directly assesses the memory capabilities needed for recalling previous decisions.

**Counterfactual thinking**  We adapt the *Iowa gambling task* (IGT) (Bechara et al., 2013) into a double choice version (DC-IGT) to assess counterfactual thinking. In traditional IGT, participants select cards from four decks with different reward-loss profiles to maximize profit. To explicitly test counterfactual thinking, our DC-IGT allows models to observe the outcomes of their initial choice and provides

an opportunity to revise their decision. By forcing models to reconsider past choices with an opportunity to 'turn back time', DC-IGT implements the mechanism of counterfactual thinking – creating an alternative to reality by considering 'what if' (Byrne, 2016).

**Belief updating**  We employ *probabilistic reversal learning task* (PRLT) (Cools et al., 2002), where participants choose between two options with differing reward probabilities. These probabilities reverse midway through the task without notice, requiring models to update their prior beliefs about the reward distributions in response to environmental changes.

**Meta-reflection**  We design a *meta-bandit task* (MBT) that extends the previous paradigm of PRLT by introducing predictable reversals. The reward probabilities (0 or 1) are reversed every **n** trials, establishing a higher-order pattern. This design tests the model's ability to recognize and anticipate the meta-structure of changes, i.e., meta-reflection, beyond mere adaptation to individual reversals.

## 3.3. Adapt Tests for Evaluating LLMs

**Weather prediction task (prediction)**  In each trial, the LLM receives two inputs: the current day's weather and a sensor state ([0,1] or [1,0]). Based on these two inputs, it predicts the next day's weather (example in AppendixA.2). The actual weather is calculated with the corresponding transition matrix:

$$T_{sensors=[1,0]} = \begin{bmatrix} p & 1-p \\ 1-p & p \end{bmatrix}$$

$$T_{sensors=[0,1]} = \begin{bmatrix} 1-p & p \\ p & 1-p \end{bmatrix}$$

For subsequent trials, LLM receives the actual weather (which becomes the current weather) and a new sensor state. We estimate the model's internal transition matrices from its predictions. Performance is evaluated using the mean absolute error (MAE) between estimated and true transition matrices.

$$Score = (1 - \text{MAE}/\text{Max}_{\text{MAE}}) * 100$$

**Wisconsin card sorting test (decision-making)**  Following (Kennedy & Nowak, 2024), we implement WCST in a text-based task with **x** trials. The matching rule (shape, color, or number) changes every **x/6** trial, allowing each rule to be tested twice. In each trial, the LLM receives a target card description (*e.g.*, 'triangle green 4') and must match it with one of four choice cards without knowing the current matching rule. The model receives feedback after each choice (example in Appendix A.3). Performance is evaluated based on overall matching accuracy.

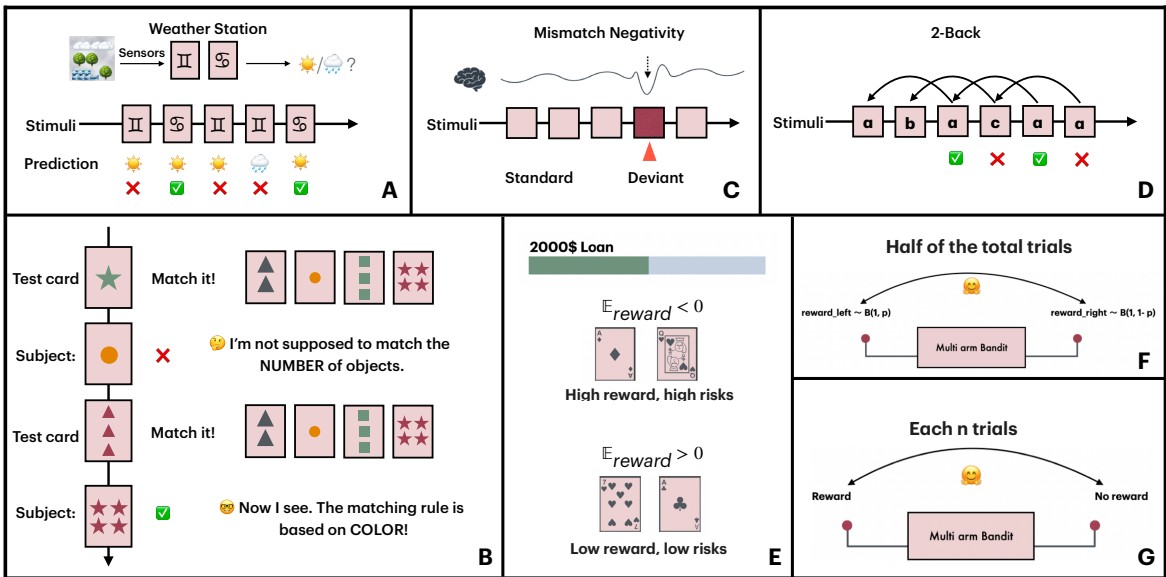

*Figure 2.* Illustration of seven cognitive tests involved in Reflection-Bench: A. weather prediction task; B: Wisconsin card sorting test; C: Oddball paradigm; D: N-back; E: Iowa gambling task; F: probabilistic reversal learning task; G: meta-bandit task.

**Oddball paradigm (perception)** We adapt the Oddball paradigm into a prompt-based task where each prompt contains eight short sentences: seven about a consistent topic (stimulus A) and one unrelated sentence (stimulus B) that disrupts the content flow (example in Appendix A.4). The LLMs are instructed to freely make comments on these prompts, allowing assessment of their spontaneous detection of contextual anomalies. We evaluate the responses using both rule-based manual scoring (0-3 scale) and an automated method based on OpenAI's text-embedding-3-large (detailed in Appendix D). The test includes 50 distinct prompt sets.

**N-back (memory)** We present a fixed sequence of letters (e.g., E, F, G, H) to the LLM one at a time. For each letter, the model must determine whether it matches the letter shown **n** steps earlier in the sequence (example in Appendix A.5). Performance is measured by response accuracy.

**Double choice Iowa gambling task (counterfactual thinking)** The DC-IGT features four card decks with fixed gains ($100, $100, $50, and $50) and potential losses ($260, $1250, $50, and $200) occurring with possibilities of $\mathbf{P_{loss}} = p_a, p_b, p_c, p_d$ . Each trial consists of two consecutive choices. The LLM first selects a deck and receives feedback on the resulting gain and potential loss. Based on this feedback, it makes a second deck selection (example in Appendix A.6). Performance is evaluated using both short-term metrics (adaptive switching behavior to avoid losses) and long-term metrics (cumulative net earnings).

**Probabilistic reversal learning task (belief updating)** In PRLT, the LLM repeatedly chooses between two options (left and right arms) with complementary reward probabilities of **p** and **1-p**. After each choice, a reward is drawn from a Bernoulli distribution according to the chosen option's probability. At the midpoint of trials, these probabilities are reversed. The LLM receives the reward feedback and makes its next choice (example in Appendix A.7). We estimate the model's internal representation of reward probabilities using a moving average (window size = 5) of its choices. Performance is quantified using the mean absolute error between estimated and true probabilities:

$$Score = (1 - \text{MAE}/\text{Max}_{\text{MAE}}) * 100$$

**Meta-bandit task (meta-reflection)** The MBT consists of 20 blocks of **n** trials each, where the LLM repeatedly chooses between two options with deterministic binary rewards, with one option yielding 1 and the other 0. After each choice, the model receives feedback on the sampled reward. The reward mappings reverse every **n** trials without notification (example in Appendix A.8). Performance is evaluated based on the model's ability to recognize this fixed reversal pattern, as indicated by sustained reward acquisition even during reversal trials.

## 4. Experiment

### 4.1. Experimental Setup

We first evaluate 16 LLMs on the Reflection-Bench:

- Large reasoning models: o1-preview, o1-mini (Ope-

Table 1. Experiment settings

| Task | Parameters | |
|------|-----------|------|
| | Easy | Hard |
| WPT | **p**=0.9 | p=0.8 |
| WCST | **x**=72 | **x**=90 |
| Oddball | NA | NA |
| N-back | **n**=2 | n=4 |
| DC-IGT | $\mathbf{P_{loss}} = \{.5, .1, .5, .1\}$ | $\{.5, .2, .5, .2\}$ |
| PRLT | **p**=0.8 | p=0.7 |
| MBT | **n**=2 | n=4 |

nAI, 2024), DeepSeek-Reasoner (R1) (DeepSeek-AI, 2024), and QwQ-32B-Preview (Qwen, 2024)

- Mainstream LLMs: GPT-4o, GPT-4o-mini (OpenAI, 2023), Claude-3.5-Sonnet, Claude-3.5-Haiku (Anthropic, 2023), Grok 2 (xAI, 2024), Gemini-2.0-flash (Google, 2024), DeepSeek-V3 (Liu et al., 2024a), and Llama-3.3-70B (Meta, 2024)

- Qwen 2.5 family: variants with 72B, 32B, 14B, and 7B (Yang et al., 2024).

All evaluations are conducted through respective model APIs. Table 1 (Easy) details the number of trials and parameter settings for each task. To ensure evaluation robustness, we repeat most tasks twice. The Oddball task, given its relatively subjective scoring, is repeated three times to minimize potential biases. We implement three response formats (see Appendix A.1): free output, direct choice generation, and zero-shot CoT (Wei et al., 2024). Due to context length constraints, large reasoning models are evaluated only through free output and direct generation, as their CoT responses often exceed maximum token limits.

To validate the long-term relevance of Reflection-Bench, we evaluate these 16 LLMs on more challenging settings (Table 1. Hard) with the direct output strategy. Oddball test is excluded because it is not parameterized. To validate the contamination-minimization of Reflection-Bench, we evaluate Centaur, which is specifically fine-tuned with human cognitive test data (Binz et al., 2024), and its base model, Llama-3.1-70B-Instruct (Grattafiori et al., 2024), on both Easy and Hard settings with the direct output strategy.

We also conducted 1 million random simulations for each applicable task, excluding the non-parameterized Oddball test and qualitative MBT. By randomly generating choices, we established chance-level thresholds at the 95th percentile of performance distributions. Performance exceeding these thresholds indicates statistically significant task performance, with higher scores reflecting a more precise inference of task parameters.

## 4.2. Experimental Results

### 4.2.1. OVERALL RESULTS

On entry-level difficulties, Claude-3.5-Sonnet achieved superior performance with both zero-shot CoT (68.78) and free output (68.23), closely followed by two large reasoning models – o1-preview (66.57) and DeepSeek-Reasoner (64.56). The performance distribution reveals a clear hierarchical structure aligned with model scaling:

- First-tier ( >60 points): seven leading models including Claude-3.5-Sonnet, o1-preview, DeepSeek-Reasoner, Gemini-2.0-flash-exp, GPT-4o, Grok-2, and DeepSeek-V3

- Second-tier (50-60 points), eight intermediate models including Llama-3.3-70B, Claude-3.5-Haiku, GPT-4o-mini, and Qwen-2.5 variants (72B/32B/14B)

- Third-tier (<50 points): Qwen-2.5-7B-Instruct, the smallest model evaluated.

Notably, all models struggle with the MBT, failing to identify the 2-step reversal pattern and making errors consistently across 20 blocks (Figure 15, 16), as discussed in Section 4.2.2. Consequently, the reported average scores exclude MBT. As detailed in Table 9, none of the 16 models achieved performances above chance-level thresholds across all five applicable tasks.

Regarding prompting strategies, free output and CoT outperformed direct generation across tasks. Strategy effectiveness varied by models and tasks (see Figure 9). o1-preview demonstrates remarkable consistency between free output and direct generation. Task sensitivity to prompting strategies ranged from high (DC-IGT & N-back) to low (PRLT & Oddball Test).

On hard settings, we observed the expected score decreases across the evaluated models, as shown in Table 9. This indicates the long-term relevance of Reflection-Bench and substantial room for LLMs development. Additionally, Centaur, the model fine-tuned with human cognitive test data, showed no improvement on Reflection-Bench compared to its base model, Llama-3.1-70B-Instruct (see Table 9). This provides empirical evidence that our parameterized design effectively minimizes data leakage concerns.

### 4.2.2. TASK-SPECIFIC ANALYSIS

**Weather Prediction Task** To reduce task complexity, we implemented highly deterministic transition probability matrices (**p**=0.9). The task design specified that given cue '[1,0]', the next day would likely remain unchanged (p=0.9), whereas cue '[0,1]' indicated a probable transition to the opposite state. Despite this simplified probability structure,

*Table 2.* Average Scores of 44 Model-Strategy Pairs on Reflection-Bench

| Model | Strategy | Score | Model | Strategy | Score |
|---|---|---|---|---|---|
| [2nd] o1-preview | Free output | 64.94 | o1-mini | Free output | **54.60** |
| | Direct generation | **66.57** | | Direct generation | 54.29 |
| [3rd] DeepSeek-Reasoner | Free output | 61.47 | QwQ-32B-preview | Free output | **58.64** |
| | Direct generation | **64.56** | | Direct generation | 48.37 |
| GPT-4o | CoT | **62.56** | GPT-4o-mini | CoT | 48.41 |
| | Free output | 52.95 | | Free output | **57.09** |
| | Direct generation | 57.29 | | Direct generation | 47.44 |
| [1st] Claude-3.5-Sonnet | CoT | **68.78** | Claude-3.5-Haiku | CoT | 57.94 |
| | Free output | 68.23 | | Free output | **58.14** |
| | Direct generation | 62.59 | | Direct generation | 55.40 |
| DeepSeek-V3 | CoT | **61.25** | Qwen-2.5-72B-Instruct | CoT | 54.79 |
| | Free output | 59.50 | | Free output | **56.74** |
| | Direct generation | 52.91 | | Direct generation | 54.26 |
| Grok-2 | CoT | 61.00 | Qwen-2.5-32B-Instruct | CoT | **51.06** |
| | Free output | **61.63** | | Free output | 48.01 |
| | Direct generation | 53.29 | | Direct generation | 49.21 |
| Gemini-2. 0-flash-exp | CoT | 61.51 | Qwen-2.5-14B-Instruct | CoT | **57.06** |
| | Free output | **63.29** | | Free output | 52.81 |
| | Direct generation | 51.13 | | Direct generation | 52.94 |
| Llama-3.3-70B | CoT | 52.44 | Qwen-2.5-7B-Instruct | CoT | 44.53 |
| | Free output | **58.92** | | Free output | **46.66** |
| | Direct generation | 53.01 | | Direct generation | 40.37 |

most models exhibited difficulty in simultaneously learning two distinct transition patterns. As illustrated in Figure 10, our analysis revealed five prediction patterns: **A**: correct predictions, **B**: predictions merely by today's weather, **C**: predictions solely by the sensor's states, **D**: partially correct predictions, and **E**: random predictions. Among all 44-model-strategy combinations (detailed in Appendix 3), only Claude-3.5-Sonnet with CoT successfully identified the transition patterns across both sessions. Four other combinations (o1-mini with free output, Claude-3.5-Sonnet with free output, and DeepSeek-Reasoner with both free output and direct generation) learned the pattern in one session. The remaining 39 combinations showed varying degrees of failure in capturing the underlying transition probabilities.

**Wisconsin Card Sorting Test**  In the WCST evaluation (Table 4), o1-preview and Claude-3.5-Sonnet achieved identical top scores (81.25), significantly outperforming Gemini-1.5-pro (64.58) in second place. A comprehensive analysis across 72 trials comprising 6 rule groups revealed distinctive behavioral patterns (Figure 11). The cross-model average accuracies for successive rule groups were 80.02% (trial 1-12, shape), 44.22% (trial 13-24, color), 26.04% (trial 25-36, number), 76.42% (trial 37-48, shape), 46.97% (trial 49-60, color), and 45.36% (trial 61-72, number), respectively. The consistently high performance in shape-based rule groups suggests that most LLMs can effectively learn and apply

specific decision rules, with Llama-3.3-70B (CoT) being a notable exception, showing persistent difficulties. However, following the initial rule group, a prevalent 'shape sink' phenomenon emerged - models defaulted to shape-based sorting despite rule changes. This 'sink' pattern persisted even in top-performing models like o1-preview and Claude-3.5-Sonnet, which showed marked performance declines during the third rule group. These findings underscore a fundamental limitation in LLMs' ability to adapt to sequential rule (or environment) changes.

**Oddball Test**  Our rule-free automated evaluation approach, leveraging text-embedding-3-large and cosine similarity metrics, demonstrated robust agreement with manual rule-based scoring (r=0.87, p=5.83e-60; detailed analysis in Appendix D. This methodology revealed distinct patterns in models' contextual anomaly detection capabilities (Table 5). Llama-3.3-70B and Claude-3.5-Sonnet exhibited superior sensitivity to deviant stimuli, followed by Gemini-2.0-flash-exp, demonstrating advanced capabilities in unexpected information detection. Conversely, Qwen-2.5-7B and o1-preview frequently overlooked contextual deviants, suggesting that contextual sensitivity may be more dependent on specific training techniques rather than model sizes.

**N-back**  In the 2-back task (Table 6), three models – o1-preview, DeepSeek-V3, and Grok-2 – achieved perfect

scores. Despite full conversation history being available during evaluation, many models struggle to determine whether the current stimulus matches the one presented 2 steps earlier. Given that the baseline accuracy for an all-no response strategy is 58%, models performing below this threshold likely demonstrate fundamental deficits in working memory capacity, specifically in maintaining and comparing information across multiple interaction turns.

**Double Choice Iowa Gambling Task**   Our evaluation of DC-IGT incorporated both short-term and long-term performance metrics. The short-term analysis (Figure 12) examined four behaviors – **insisting** with gain, **insisting** with risk, unnecessary **switch**, and loss-avoiding **switch**. Among these, insisting with gain emerged as the dominant behavior. Some models, notably GPT-4o-mini with free output, demonstrated optimal decision-making through balanced deployment of insisting with gain and loss-avoiding switch strategies. In contrast, Claude-3.5-Sonnet with free output exhibited frequent risk-insisting behavior, resulting in poor overall performance (Table 7). To differentiate between strategic adaptation and chance-based success, we implemented long-term metrics focusing on expected reward learning across four card decks. This approach specifically evaluated whether forced counterfactual thinking enhanced strategic optimization. The long-term analysis (Figure 13) tracked average coverage changes across model-strategy pairs, highlighting eight representative cases. Gemini-2.0-flash-exp excelled in both temporal dimensions, achieving the highest overall score and demonstrating robust counterfactual reasoning abilities.

**Probabilistic Reversal Learning Task**   Beyond aggregate performance metrics (Table 8), our behavioral analysis (Figure 14) revealed distinct patterns in belief updating across different model-strategy pairs. All five models scoring below 50 points exhibit obvious rigid beliefs, failing to adapt to the probability reversal after trial 20. Models achieving scores above 60 points demonstrated adaptive belief updating capabilities. A prevalent 'win-stay-lose-switch' strategy emerged across most models, suggesting a capacity for local adaptation. We further explored whether LLMs can develop a higher-order understanding of environmental structure through MBT.

**Meta Bandit Task**   In MBT, a striking limitation emerged: no model successfully recognized the 2-step-reversal pattern (rewards alternating between options every two trials) across 20 reversal blocks (Figure 15, 16). Instead, all models exhibited either periodic or irregular errors. While most models defaulted to the 'win-stay-lose-switch' strategy that proved effective in PRLT, this reactive approach proved insufficient for capturing the global temporal pattern in MBT. This inability to identify even simple temporal patterns reveals

that current LLMs struggle to abstract global patterns from experience and leverage them for future decision-making. This constraint represents a critical limitation for LLMs as autonomous agents.

### 4.3. Effects of Prompting Strategies

Figure 17 presents a comparative analysis of three prompting strategies. Zero-shot CoT and free output achieved similar overall performances, both surpassing direct generation. Breaking down the impacts by task reveals that zero-shot CoT exhibited notable advantages in WCST and 2-back tasks while showing reduced effectiveness in DC-IGT. The impact of zero-shot CoT varied across models as well: while it enhanced GPT-4o's performance, it degraded the performance of models like GPT-4o-mini to levels comparable with direct generation. These findings suggest that the efficacy of zero-shot CoT is contingent on a model's capacity to leverage structured reasoning prompts. The mixed effects can be attributed to the inherent integration of CoT-like mechanisms in many LLMs, potentially limiting the additional benefits of explicit zero-shot CoT. In the context of Reflection-Bench, these results indicate that autonomous agents operating in complex environments require task-specific cognitive strategies - some tasks benefit from rapid, intuitive responses where extended deliberation may be counterproductive (Liu et al., 2024b), while others demand systematic, structured reasoning(Sprague et al., 2024). This pattern parallels human cognition theories, where adaptive strategy selection is crucial for effective problem-solving across diverse scenarios.

## 5. Discussion

**Key Findings**   The hierarchical performance pattern indicates the Reflection-Bench's discriminative power. The supplementary evaluations verify Reflection-Bench's long-term relevance and resilience against data contamination. On entry-level configurations, top-tier models achieve fair performance in most individual tasks except MBT, demonstrating a basic level of epistemic agency. However, detailed behavioral analyses also reveal significant limitations of LLMs' capabilities, especially in prediction, decision-making, and meta-reflection. Illustrated by performances in PRLT and MBT, current LLMs, without global structural understanding and meta-cognitive regulation, are largely driven by short-sighted local adaptation patterns. The effectiveness of prompting strategies varies substantially across both tasks and models, suggesting that the interplay between task characteristics and interaction strategies significantly impacts agent performance. Notably, no model-strategy combination maintains consistently good performance across all tasks. Altogether, our evaluation reveals both the emergence and limitations of epistemic agency in current LLMs.

**Implications** Our findings provide several implications for developing more robust epistemic agency in language models. The substantial weaknesses observed in meta-reflection suggest future research needs to focus on innovations that enhance meta-cognitive capabilities (Scholten et al., 2024), which would benefit from actively regulating thought processes and contributing to rational reasoning, improved learning, and reliable decision-making (Griffiths, 2020; Boureau et al., 2015). Develop prompts or fine-tuning strategies that encourage dynamic shifts between rapid "intuition" and more deliberative "reflection" based on situational demands. While long-form CoT reasoning has gained prominence in advanced models such as OpenAI's o1, concerns about its drawbacks have emerged (Chen et al., 2024b; Shaikh et al., 2023; Liu et al., 2024b), highlighting the importance of situation-appropriate transitions. Beyond enhancing individual capabilities, a key challenge lies in fostering organic coordination between different cognitive processes. As discussed in Section 3.1, such coordinated development is essential for achieving genuine epistemic agency across diverse real-world scenarios. Current research provides limited insights into how different cognitive components interact within LLMs. Understanding these internal dynamics, particularly how different aspects of epistemic agency emerge and coordinate, could enhance both model interpretability and trustworthiness.

**Limitations** First, we emphasize that Reflection-Bench specifically examines epistemic agency as a cognitive-level characteristic in base LLMs, rather than evaluating specialized agent architectures with additional modules. This focused scope allows us to isolate and assess LLMs independent of specific external components or application scenarios. But it is unclear how epistemic agency might manifest in systems where multiple LLMs collaborate as an agent core. Second, Reflection-Bench involves seven cognitive tests adapted to evaluate LLMs, but the ecological validity of these adapted tasks may require further investigation. Third, while Reflection-Bench currently focuses on linguistic interaction as a medium for studying LLMs' epistemic agency, we acknowledge the rich opportunities presented by emerging multi-modal LLMs and embodied intelligence systems. Future iterations could expand to include enriched sensory or visual contexts to explore how epistemic agency manifests across different modalities. Additionally, to establish reliable measurements, our tasks are designed with structured episodes and clear feedback signals. Future work could complement these controlled assessments by exploring epistemic agency in more naturalistic contexts, such as game (Allen et al., 2024) and open-ended dialogues with shifting goals or ambiguous instructions. Although the parameterized design of Reflection-Bench efficiently reduces the risks of data leakage, future work could develop regular updating designs similar to LiveBench (White et al., 2024),

which could further address contamination concerns. Overall, while our current benchmark provides a novel framework for evaluating epistemic agency in LLMs, these directions for future work highlight the rich landscape of possibilities for a deeper understanding of artificial agency.

## 6. Conclusion

This paper presents Reflection-Bench, a novel benchmark for evaluating the intrinsic epistemic agency of Large Language Models through the lens of cognitive psychology. By examining the cognitive process of efficient agent-environment interaction, we decompose epistemic agency into seven interrelated dimensions, including prediction, decision-making, perception, memory, counterfactual thinking, belief updating, and meta-reflection. Building on established paradigms in cognitive psychology, we select and adapt seven cognitive tests for Reflection-Bench. The parameterized design ensures minimization of potential data contamination and maintains its long-term relevance for future model evaluation.

Our comprehensive evaluation of 16 models using three prompting strategies on entry-level difficulties reveals a clear performance hierarchy and demonstrates that state-of-the-art models exhibit a preliminary epistemic agency. The supplementary hard-parameter evaluations demonstrated expected performance decreases across all models, verifying Reflection-Bench's long-term relevance and substantial headroom for model improvement. To validate the benchmark's resilience against data contamination, we compared Centaur, a model specifically fine-tuned with human cognitive test data, against its base model, finding no significant performance improvement, confirming the effectiveness of our contamination-minimization design. The detailed behavioral analyses demonstrate critical limitations in current models, particularly in prediction, decision-making, and meta-reflection. The varying effectiveness of different prompting strategies across tasks and models suggests that successful epistemic agency requires dynamic cognitive strategies rather than fixed approaches.

These findings point to several critical directions for advancing epistemic agency in LLMs, such as enhancing meta-cognition, developing mechanisms for dynamic shifts between intuitive and deliberative reasoning, and fostering organic coordination among cognitive capabilities. Future work exploring agency manifestation in multi-LLM collaboration, multi-modal interactions, embodied systems, and more naturalistic contexts could yield more insights into our current framework. As AI systems continue to evolve, understanding and enhancing their epistemic agency will be increasingly crucial for developing more reliable LLMs-based agents.

## Acknowledgments

This paper is supported by Shanghai Artificial Intelligence Laboratory. We appreciate Kexin Huang from Fudan University for her assistance in improving the visual representations.

## Impact Statement

This work introduces Reflection-Bench, a novel cognitive-psychology-inspired framework that systematically evaluates large language models' intrinsic epistemic agency. By revealing both emerging capabilities and limitations in current models' epistemic agency, this research provides crucial insights for enhancing LLMs' epistemic agency towards more reliable AI agents.

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

# A. System Prompts and Task Examples

## A.1. Strategy Prompts

**Chain of Thought**: user prompt + ''let's think step by step."

**Direct Generation**: user prompt + "respond only with your choice directly without outputting any other information or analysis."

**Free output**: user prompt + ""

## A.2. Weather Prediction Task

**System Prompt**   You are an expert forecaster working in a weather station. There are two devices collecting data from nature. Your task is to predict tomorrow's weather based on (1) today's weather and (2) the current states of four sensor devices in the weather station. Here's how the task works: 1. There are two devices, each represented by either 0 (inactive) or 1 (active). 2. The device states will be given to you in the format [d1,d2], where each d is either 0 or 1; 3. Based on these device states and today's weather, you need to predict whether tomorrow's weather will be sunny or rainy. 4. After your prediction, I will inform you of the actual weather outcome. 5. We will repeat this process multiple times, and you should try to improve your predictions based on the feedback. At each time, make your prediction of the next day's weather ('sunny' or 'rainy').

The task example is illustrated in Figure 3.

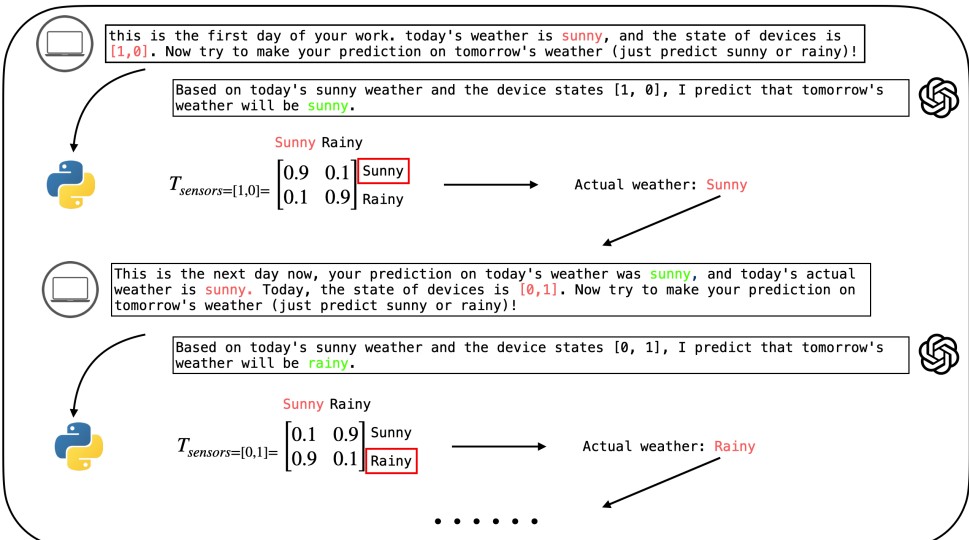

*Figure 3.* Example of Weather Prediction Task

## A.3. Wisconsin Card Sorting Test

**System prompt**   You are performing an interesting Task. In this task, you have four cards on your desk, that is, 'triangle red 1', 'cross green 2', 'circle yellow 1', and 'star blue 4'. The three word/figure represent (1) the type of shape, i.e. triangle, cross, circle, or star, (2) the color of the shape, i.e. red, green, yellow, or blue, and (3) the number of the shape, i.e., 1, 2, 3, or 4, respectively. At each trial, you will be presented with a testing card. You should point out which card on your desk matches the testing card. I will not tell you the matching rule, but only provide feedback if your choice was right or wrong. Your primary goal is to strive to maximize your accuracy rate. Respond with your option ('triangle red 1', 'cross green 2', 'circle yellow 1', or 'star blue 4'). Keep performing the task until the end of the test.

The task example is illustrated in Figure 4.

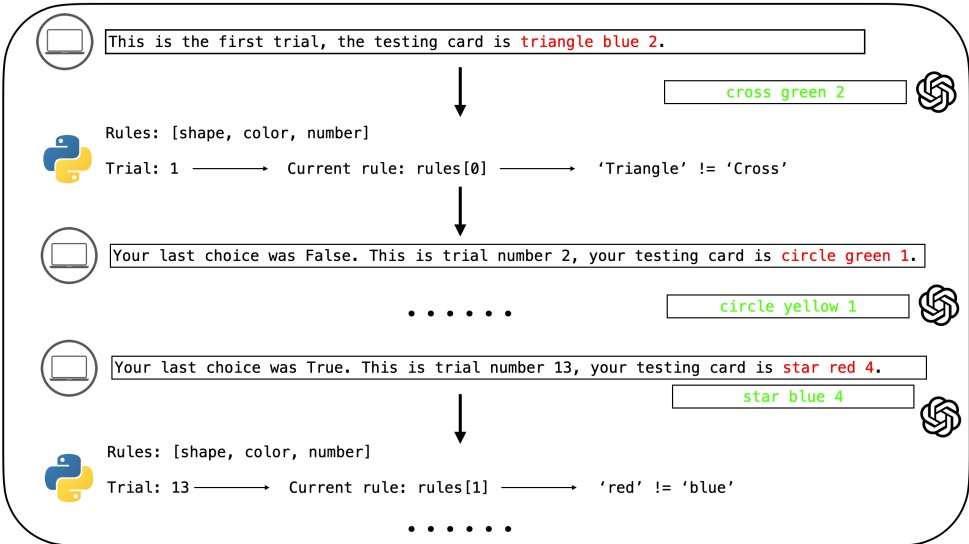

*Figure 4.* Example of Wisconsin Card Sorting Test

## A.4. Oddball Test

**System prompt**    You are playing a game and will be presented with a sequence of sentences about a specific topic. Just make some short comments on the material.

Task examples can be found in Appendix D.

## A.5. N-back (2-back)

**System prompt**    You are playing a game. I will give you a series of characters in sequence, showing only one at a time. Your task is to determine whether the current character is the same as the character 2 steps before. If the current character is the same as the character 2 steps before, answer Yes. If the current character is different from the character 2 steps before, answer No. For the first 2 steps, since there aren't enough preceding characters for comparison, answer Not Available.

The task example is illustrated in Figure 5.

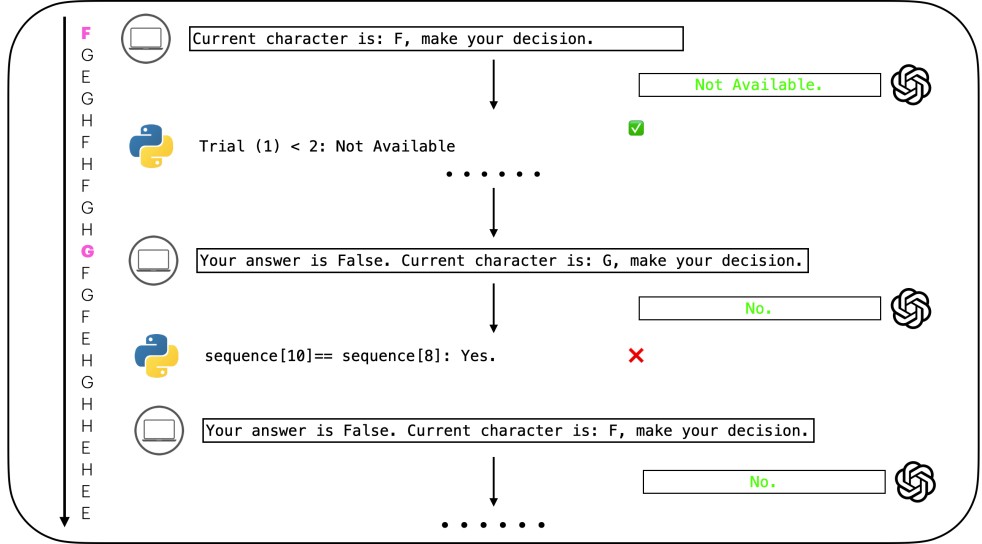

*Figure 5.* Example of N-back task (2-back)

### A.6. Double Choice Iowa Gambling Task

**System prompt**    You are participating in an interesting game. There are four decks of cards in front of you, labeled 'AAA', 'BBB', 'CCC', and 'DDD'. You start with $2000 in your account. Your goal is to maximize your account assets by making a series of choices from these decks. For each trial: 1. Choose one deck ('AAA', 'BBB', 'CCC', or 'DDD'). 2. You will receive feedback on the outcome of your choice, including any gains and losses. 3. After receiving the feedback, you have one opportunity to reconsider your initial decision; you can either stick with your original choice or make a new choice. 4. Your final choice would determine your actual gain or loss for the trial. Respond with your deck choice (AAA, BBB, CCC, or DDD) for each choice. keep making choices until instructed to stop.

The task example is illustrated in Figure 6.

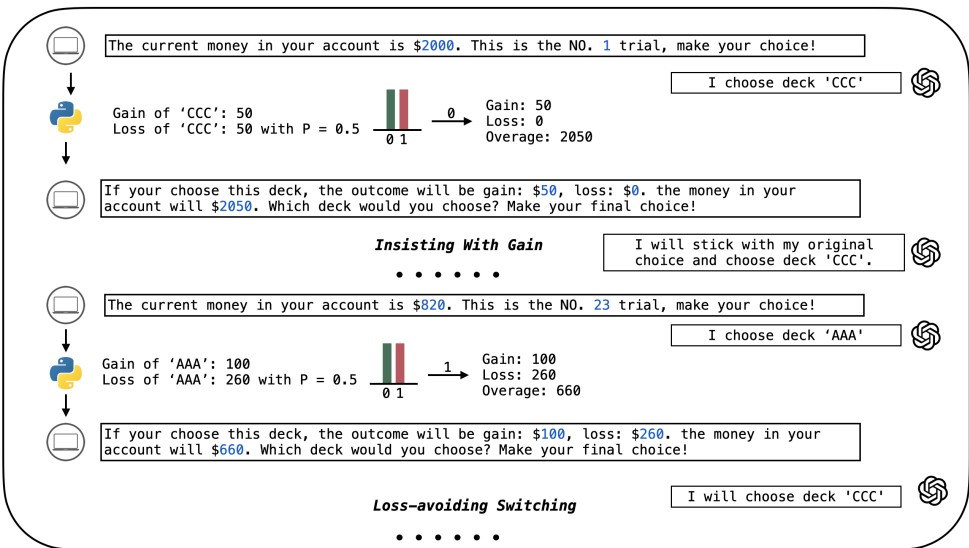

*Figure 6.* Example of Double Choice Iowa Gambling Task

### A.7. Probabilistic Reversal Learning Task

**System prompt**    You are playing a two-arm bandit game. Each time you need to choose between the right arm and the left arm. You will receive feedback (0 or 1) based on your choice. Your goal is to maximize the total reward. Keep performing the task until the end of the test.

The task example is illustrated in Figure 7.

### A.8. Meta Bandit Task

**System prompt**    You are playing a two-arm bandit game. Each time, there is one rewarding arm, and you need to choose between the right arm and the left arm. You will receive feedback (0 or 1) based on your choice. Your goal is to maximize the total reward. Respond with your choice (right arm or left arm). Keep performing the task until the end of the test.

The task example is illustrated in Figure 8.

## B. Raw Results

In this appendix, we provide detailed raw scores of all model-strategy pairs in the Weather Prediction Task (Table 3), Wisconsin Card Sorting Task (Table 4), Oddball Test (Table 5), N-back (Table 6), Double Choice Iowa Gambling Task (Table 7), and Probabilistic Reversal Learning Task (Table 8).

**P.S.**    Illustrated in Figure 15 & 16, all models failed to recognize the 2-step reversal pattern and performed insisted mistakes across the 20 blocks. Therefore, we only conduct behavioral analysis of the Meta Bandit Task.

*Table 3.* Scores of Weather Prediction Task

| Model | Strategy | Score | Model | Strategy | Score |
|---|---|---|---|---|---|
| o1-preview | Free output | 58.92 | o1-mini | Free output | 68.81 |
| | Direct generation | 60.28 | | Direct generation | 44.85 |
| DeepSeek-Reasoner | Free output | 71.78 | QwQ-32B-preview | Free output | 41.78 |
| | Direct generation | 81.03 | | Direct generation | 34.78 |
| GPT-4o | CoT | 43.92 | GPT-4o-mini | CoT | 43.92 |
| | Free output | 28.33 | | Free output | 28.33 |
| | Direct generation | 60.02 | | Direct generation | 60.02 |
| Claude-3.5-Sonnet | CoT | **85.36** | Claude-3.5-Haiku | CoT | 33.61 |
| | Free output | 75.09 | | Free output | 58.07 |
| | Direct generation | 50.25 | | Direct generation | 48.11 |
| DeepSeek-V3 | CoT | 49.87 | Qwen-2.5-72B-Instruct | CoT | 38.48 |
| | Free output | 46.48 | | Free output | 41.16 |
| | Direct generation | 47.36 | | Direct generation | 43.53 |
| Grok-2 | CoT | 33.97 | Qwen-2.5-32B-Instruct | CoT | 44.33 |
| | Free output | 50.48 | | Free output | 32.00 |
| | Direct generation | 34.14 | | Direct generation | 36.72 |
| Gemini-2. 0-flash-exp | CoT | 44.35 | Qwen-2.5-14B-Instruct | CoT | 49.50 |
| | Free output | 57.13 | | Free output | 52.75 |
| | Direct generation | 42.58 | | Direct generation | 50.33 |
| Llama-3.3-70B | CoT | 51.00 | Qwen-2.5-7B-Instruct | CoT | 43.23 |
| | Free output | 42.10 | | Free output | 44.35 |
| | Direct generation | 34.44 | | Direct generation | 38.75 |

*Table 4.* Scores of Wisconsin Card Sorting Test

| Model | Strategy | Score | Model | Strategy | Score |
|---|---|---|---|---|---|
| o1-preview | Free output | 80.56 | o1-mini | Free output | 54.17 |
| | Direct generation | **81.25** | | Direct generation | 57.64 |
| DeepSeek-Reasoner | Free output | 56.25 | QwQ-32B-preview | Free output | 54.17 |
| | Direct generation | 49.31 | | Direct generation | 43.75 |
| GPT-4o | CoT | 68.75 | GPT-4o-mini | CoT | 55.56 |
| | Free output | 54.17 | | Free output | 54.17 |
| | Direct generation | 48.61 | | Direct generation | 43.75 |
| Claude-3.5-Sonnet | CoT | **81.25** | Claude-3.5-Haiku | CoT | 61.11 |
| | Free output | 79.86 | | Free output | 51.39 |
| | Direct generation | 47.92 | | Direct generation | 38.19 |
| DeepSeek-V3 | CoT | 55.56 | Qwen-2.5-72B-Instruct | CoT | 58.33 |
| | Free output | 53.47 | | Free output | 52.78 |
| | Direct generation | 47.22 | | Direct generation | 43.06 |
| Grok-2 | CoT | 61.81 | Qwen-2.5-32B-Instruct | CoT | 46.53 |
| | Free output | 50.00 | | Free output | 48.61 |
| | Direct generation | 49.31 | | Direct generation | 43.75 |
| Gemini-2. 0-flash-exp | CoT | 64.58 | Qwen-2.5-14B-Instruct | CoT | 61.11 |
| | Free output | 50.00 | | Free output | 47.92 |
| | Direct generation | 43.06 | | Direct generation | 41.67 |
| Llama-3.3-70B | CoT | 32.64 | Qwen-2.5-7B-Instruct | CoT | 48.61 |
| | Free output | 52.08 | | Free output | 40.97 |
| | Direct generation | 43.06 | | Direct generation | 41.67 |

*Table 5.* Scores of Oddball Test

| Model | Strategy | Score | Model | Strategy | Score |
|---|---|---|---|---|---|
| o1-preview | Free output | 27.20 | o1-mini | Free output | 32.34 |
| | Direct generation | 25.10 | | Direct generation | 30.90 |
| DeepSeek-Reasoner | Free output | 45.05 | QwQ-32B-preview | Free output | 47.33 |
| | Direct generation | 40.46 | | Direct generation | 42.34 |
| GPT-4o | CoT | 33.43 | GPT-4o-mini | CoT | 33.53 |
| | Free output | 34.28 | | Free output | 33.25 |
| | Direct generation | 28.57 | | Direct generation | 28.05 |
| Claude-3.5-Sonnet | CoT | 52.90 | Claude-3.5-Haiku | CoT | 45.26 |
| | Free output | 55.23 | | Free output | 44.23 |
| | Direct generation | 52.01 | | Direct generation | 38.19 |
| DeepSeek-V3 | CoT | 41.78 | Qwen-2.5-72B-Instruct | CoT | 37.35 |
| | Free output | 46.57 | | Free output | 38.94 |
| | Direct generation | 37.52 | | Direct generation | 31.67 |
| Grok-2 | CoT | 46.83 | Qwen-2.5-32B-Instruct | CoT | 35.16 |
| | Free output | 45.64 | | Free output | 33.53 |
| | Direct generation | 42.07 | | Direct generation | 34.87 |
| Gemini-2. 0-flash-exp | CoT | 53.53 | Qwen-2.5-14B-Instruct | CoT | 48.36 |
| | Free output | 47.54 | | Free output | 49.11 |
| | Direct generation | 36.01 | | Direct generation | 39.48 |
| Llama-3.3-70B | CoT | 59.25 | Qwen-2.5-7B-Instruct | CoT | 30.01 |
| | Free output | **59.96** | | Free output | 25.86 |
| | Direct generation | 59.19 | | Direct generation | 21.79 |

*Table 6.* Scores of N-back (2-back)

| Model | Strategy | Score | Model | Strategy | Score |
|---|---|---|---|---|---|
| o1-preview | Free output | 89.42 | o1-mini | Free output | 85.58 |
| | Direct generation | **100.00** | | Direct generation | 86.54 |
| DeepSeek-Reasoner | Free output | 94.23 | QwQ-32B-preview | Free output | 75.00 |
| | Direct generation | 79.81 | | Direct generation | 56.73 |
| GPT-4o | CoT | 95.19 | GPT-4o-mini | CoT | 60.58 |
| | Free output | 73.08 | | Free output | 61.54 |
| | Direct generation | 67.31 | | Direct generation | 54.81 |
| Claude-3.5-Sonnet | CoT | 90.38 | Claude-3.5-Haiku | CoT | 79.81 |
| | Free output | 93.26 | | Free output | 61.54 |
| | Direct generation | 90.38 | | Direct generation | 64.42 |
| DeepSeek-V3 | CoT | **100.00** | Qwen-2.5-72B-Instruct | CoT | 61.54 |
| | Free output | 83.65 | | Free output | 77.88 |
| | Direct generation | 64.42 | | Direct generation | 66.35 |
| Grok-2 | CoT | **100.00** | Qwen-2.5-32B-Instruct | CoT | 57.69 |
| | Free output | 82.69 | | Free output | 65.38 |
| | Direct generation | 69.23 | | Direct generation | 61.54 |
| Gemini-2. 0-flash-exp | CoT | 76.92 | Qwen-2.5-14B-Instruct | CoT | 80.77 |
| | Free output | 71.15 | | Free output | 50.96 |
| | Direct generation | 58.65 | | Direct generation | 59.62 |
| Llama-3.3-70B | CoT | 59.62 | Qwen-2.5-7B-Instruct | CoT | 43.27 |
| | Free output | 66.35 | | Free output | 53.85 |
| | Direct generation | 65.39 | | Direct generation | 47.12 |

*Table 7.* Scores of Double Choice Iowa Gambling Test

| Model | Strategy | Score | Model | Strategy | Score |
|---|---|---|---|---|---|
| o1-preview | Free output | 61.30 | o1-mini | Free output | 40.23 |
| | Direct generation | 58.41 | | Direct generation | 54.68 |
| DeepSeek-Reasoner | Free output | 41.93 | QwQ-32B-preview | Free output | 63.59 |
| | Direct generation | 75.54 | | Direct generation | 64.21 |
| GPT-4o | CoT | 61.80 | GPT-4o-mini | CoT | 34.96 |
| | Free output | 55.96 | | Free output | 74.86 |
| | Direct generation | 64.02 | | Direct generation | 42.54 |
| Claude-3.5-Sonnet | CoT | 43.96 | Claude-3.5-Haiku | CoT | 60.45 |
| | Free output | 36.98 | | Free output | 64.87 |
| | Direct generation | 71.60 | | Direct generation | 66.81 |
| DeepSeek-V3 | CoT | 71.43 | Qwen-2.5-72B-Instruct | CoT | 66.91 |
| | Free output | 57.40 | | Free output | 63.40 |
| | Direct generation | 49.25 | | Direct generation | 62.59 |
| Grok-2 | CoT | 54.89 | Qwen-2.5-32B-Instruct | CoT | 52.68 |
| | Free output | 67.93 | | Free output | 28.72 |
| | Direct generation | 51.43 | | Direct generation | 42.75 |
| Gemini-2. 0-flash-exp | CoT | 61.17 | Qwen-2.5-14B-Instruct | CoT | 25.89 |
| | Free output | **80.82** | | Free output | 45.58 |
| | Direct generation | 62.81 | | Direct generation | 57.28 |
| Llama-3.3-70B | CoT | 51.43 | Qwen-2.5-7B-Instruct | CoT | 38.12 |
| | Free output | 59.19 | | Free output | 45.19 |
| | Direct generation | 46.92 | | Direct generation | 36.95 |

*Table 8.* Scores of Probabilistic Reversal Learning Task

| Model | Strategy | Score | Model | Strategy | Score |
|---|---|---|---|---|---|
| o1-preview | Free output | 71.87 | o1-mini | Free output | 46.12 |
| | Direct generation | 74.37 | | Direct generation | 49.45 |
| DeepSeek-Reasoner | Free output | 59.54 | QwQ-32B-preview | Free output | 69.74 |
| | Direct generation | 61.32 | | Direct generation | 45.28 |
| GPT-4o | CoT | 70.66 | GPT-4o-mini | CoT | 68.20 |
| | Free output | 71.74 | | Free output | 74.00 |
| | Direct generation | 73.78 | | Direct generation | 72.20 |
| Claude-3.5-Sonnet | CoT | 57.63 | Claude-3.5-Haiku | CoT | 67.51 |
| | Free output | 66.34 | | Free output | 69.70 |
| | Direct generation | 63.38 | | Direct generation | 76.59 |
| DeepSeek-V3 | CoT | 48.86 | Qwen-2.5-72B-Instruct | CoT | 66.02 |
| | Free output | 69.25 | | Free output | 65.40 |
| | Direct generation | 70.60 | | Direct generation | 76.34 |
| Grok-2 | CoT | 68.34 | Qwen-2.5-32B-Instruct | CoT | 69.98 |
| | Free output | 73.06 | | Free output | **78.86** |
| | Direct generation | 72.82 | | Direct generation | 72.99 |
| Gemini-2. 0-flash-exp | CoT | 68.08 | Qwen-2.5-14B-Instruct | CoT | 75.41 |
| | Free output | 73.12 | | Free output | 70.48 |
| | Direct generation | 62.90 | | Direct generation | 67.54 |
| Llama-3.3-70B | CoT | 60.08 | Qwen-2.5-7B-Instruct | CoT | 63.18 |
| | Free output | 72.73 | | Free output | 69.56 |
| | Direct generation | 68.48 | | Direct generation | 49.44 |

*Table 9.* Performance of 18 Models on Five Tests in Reflection-Bench Across Different Prompting Strategies and Difficulty Levels

| Model | | | WPT | WCST | N-back | DC-IGT | PRLT | Overall |
|---|---|---|---|---|---|---|---|---|
| **Chance Level** | Easy | / | 51.84 (69.95) | 61.80 (66.67) | 48.07 (59.62) | 2.51 (16.48) | 56.47 (67.87) | 40.95 |
| (95% threshold) | Hard | / | 55.80 (75.94) | 55.28 (60.00) | 46.17 (57.69) | 11.48 (19.67) | 57.72 (66.64) | 41.77 |
| o1-preview | Easy | Free | 58.92 | 80.56 | 89.42 | 61.30 | 71.87 | 72.41 |
| | | Direct | 60.28 | 81.25 | 100.00 | 58.41 | 74.37 | 74.86 |
| | Hard | Direct | 52.38 | 41.11 | 42.31 | 47.57 | 64.92 | 49.66 |
| DeepSeek-Reasoner | Easy | Free | 71.78 | 56.25 | 94.23 | 41.93 | 59.54 | 64.75 |
| | | Direct | 81.03 | 49.31 | 79.81 | 75.54 | 61.32 | 69.40 |
| | Hard | Direct | 54.49 | 43.33 | 75.96 | 60.02 | 51.29 | 57.02 |
| o1-mini | Easy | Free | 68.81 | 54.17 | 85.58 | 40.23 | 46.12 | 58.98 |
| | | Direct | 44.85 | 57.64 | 86.54 | 54.68 | 49.45 | 58.63 |
| | Hard | Direct | 29.14 | 41.67 | 51.92 | 14.93 | 62.62 | 40.06 |
| QwQ-32B-preview | Easy | Free | 41.78 | 54.17 | 75.00 | 63.59 | 69.74 | 60.86 |
| | | Direct | 34.78 | 43.75 | 56.73 | 64.21 | 45.28 | 48.95 |
| | Hard | Direct | 29.13 | 43.33 | 59.61 | 27.50 | 54.46 | 42.81 |
| GPT-4o | | CoT | 43.92 | 68.75 | 95.19 | 61.80 | 70.66 | 68.06 |
| | Easy | Free | 28.33 | 54.17 | 73.08 | 55.96 | 71.74 | 56.66 |
| | | Direct | 60.02 | 48.61 | 67.31 | 64.02 | 73.48 | 62.69 |
| | Hard | Direct | 40.08 | 44.44 | 59.62 | 44.27 | 64.25 | 50.53 |
| Claude-3.5-Sonnet | | CoT | 85.36 | 81.25 | 90.38 | 43.96 | 57.63 | 71.72 |
| | Easy | Free | 75.09 | 79.86 | 93.26 | 36.98 | 66.34 | 70.31 |
| | | Direct | 50.25 | 47.92 | 90.38 | 71.60 | 63.38 | 64.71 |
| | Hard | Direct | 59.51 | 43.88 | 72.12 | 30.82 | 55.32 | 52.33 |
| DeepSeek-V3 | | CoT | 49.87 | 55.56 | 100.00 | 71.43 | 48.86 | 65.14 |
| | Easy | Free | 46.48 | 53.47 | 83.65 | 57.40 | 69.25 | 62.05 |
| | | Direct | 47.36 | 47.22 | 64.42 | 49.25 | 70.60 | 55.77 |
| | Hard | Direct | 31.78 | 43.33 | 59.62 | 55.85 | 66.05 | 52.33 |
| Grok2 | | CoT | 33.97 | 61.81 | 100.00 | 54.89 | 48.86 | 59.91 |
| | Easy | Free | 50.48 | 50.00 | 82.69 | 67.93 | 69.25 | 64.07 |
| | | Direct | 34.14 | 49.31 | 69.23 | 51.43 | 72.82 | 55.39 |
| | Hard | Direct | 36.93 | 50.00 | 67.30 | 39.52 | 64.68 | 51.69 |
| Gemini-2.0-flash-exp | | CoT | 44.35 | 64.58 | 76.92 | 61.17 | 68.08 | 63.16 |
| | Easy | Free | 57.13 | 50.00 | 71.15 | 80.82 | 73.21 | 66.46 |
| | | Direct | 42.58 | 43.06 | 58.65 | 62.81 | 62.90 | 54.00 |
| | Hard | Direct | 40.15 | 43.33 | 77.88 | 28.23 | 67.26 | 51.37 |
| Llama-3.3-70B-Instruct | | CoT | 51.00 | 32.64 | 59.62 | 51.43 | 60.08 | 50.96 |
| | Easy | Free | 42.10 | 52.08 | 66.35 | 59.19 | 72.73 | 58.49 |
| | | Direct | 34.44 | 43.06 | 65.39 | 46.92 | 68.48 | 51.66 |
| | Hard | Direct | 44.93 | 44.44 | 59.62 | 30.15 | 58.88 | 47.60 |
| GPT-4o-mini | | CoT | 43.92 | 55.56 | 60.58 | 34.96 | 68.20 | 52.64 |
| | Easy | Free | 28.33 | 54.17 | 61.54 | 74.86 | 74.00 | 58.58 |
| | | Direct | 60.02 | 43.75 | 54.81 | 42.54 | 72.20 | 54.66 |
| | Hard | Direct | 37.00 | 40.00 | 53.85 | 42.75 | 61.93 | 47.11 |
| Claude-3.5-Haiku | | CoT | 33.61 | 61.11 | 79.81 | 60.45 | 67.51 | 60.50 |
| | Easy | Free | 58.07 | 51.39 | 61.54 | 64.87 | 69.70 | 61.11 |
| | | Direct | 48.11 | 38.19 | 64.42 | 66.81 | 76.59 | 58.82 |
| | Hard | Direct | 47.40 | 35.56 | 75.00 | 68.38 | 62.93 | 57.85 |
| Qwen-2.5-72B-Instruct | | CoT | 38.48 | 58.33 | 61.54 | 66.91 | 66.02 | 58.26 |
| | Easy | Free | 41.16 | 52.78 | 77.88 | 63.40 | 65.40 | 60.12 |
| | | Direct | 43.53 | 43.06 | 66.35 | 62.59 | 76.34 | 58.37 |
| | Hard | Direct | 30.71 | 43.33 | 58.65 | 45.43 | 63.81 | 48.39 |
| Qwen-2.5-32B-Instruct | | CoT | 44.33 | 46.53 | 57.69 | 52.68 | 69.98 | 54.24 |
| | Easy | Free | 32.00 | 48.61 | 65.39 | 28.72 | 78.86 | 50.72 |
| | | Direct | 36.72 | 43.75 | 61.54 | 42.75 | 72.99 | 51.55 |
| | Hard | Direct | 36.80 | 44.44 | 66.35 | 29.35 | 54.42 | 46.27 |
| Qwen-2.5-14B-Instruct | | CoT | 49.50 | 61.11 | 80.77 | 25.89 | 75.41 | 58.54 |
| | Easy | Free | 52.75 | 47.92 | 50.96 | 45.58 | 70.48 | 55.29 |
| | | Direct | 50.33 | 41.67 | 59.62 | 57.28 | 67.54 | 55.29 |
| | Hard | Direct | 48.44 | 44.44 | 59.62 | 69.77 | 59.87 | 56.43 |
| Qwen-2.5-7B-Instruct | | CoT | 43.23 | 48.61 | 43.27 | 38.12 | 63.18 | 47.28 |
| | Easy | Free | 44.35 | 40.97 | 53.85 | 45.58 | 69.56 | 50.86 |
| | | Direct | 38.75 | 41.67 | 47.12 | 36.95 | 49.44 | 42.79 |
| | Hard | Direct | 26.61 | 42.77 | 57.69 | 27.54 | 45.97 | 40.12 |
| Centaur | Easy | Direct | 47.51 | 31.94 | 50.96 | 37.31 | 71.68 | 46.34 |
| | Hard | Direct | 46.16 | 52.78 | 46.15 | 36.03 | 39.89 | 44.20 |
| Llama-3.1-70B-Instruct | Easy | Direct | 34.58 | 54.86 | 67.31 | 54.81 | 64.28 | 55.17 |
| | Hard | Direct | 40.58 | 46.11 | 59.62 | 48.03 | 62.43 | 51.35 |

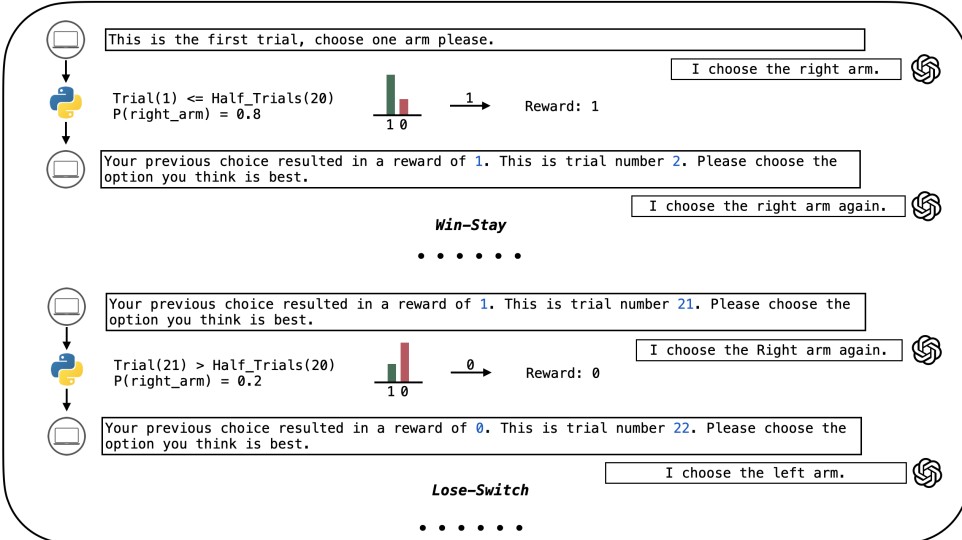

*Figure 7.* Example of Probabilistic Reversal Learning Task

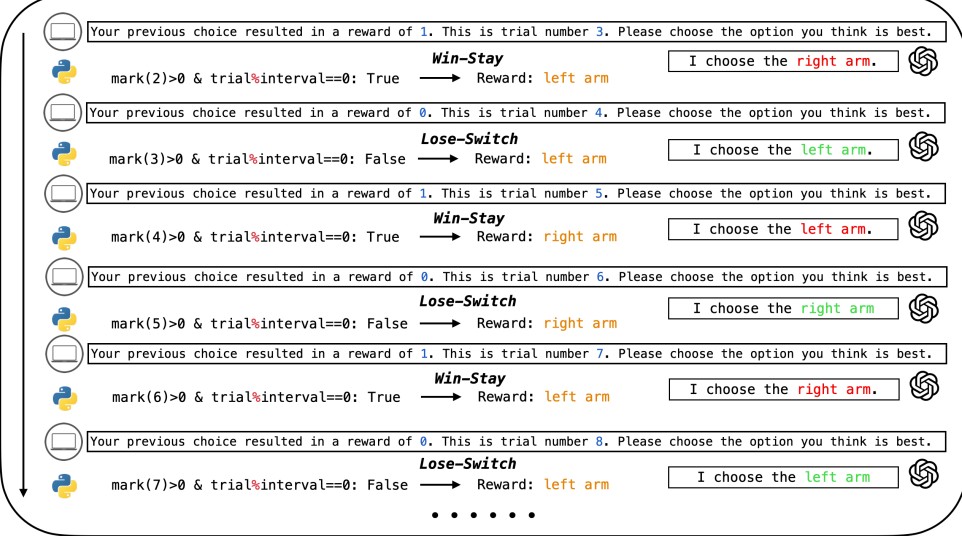

*Figure 8.* Example of Meta Bandit Task from GPT-4o, insisting the 'win-stay-lose-switch' strategy while failing to identify the global patterns.

## C. Secondary Analysis

This section provides an in-depth supplementary analysis focusing on the performance patterns of models across tasks, strategies, and sessions. Key visualizations include:

- **Model x Task x Strategy Variance (Figure 9):** Highlights the variability in model performance across different task-strategy combinations.

- **Patterns in Weather Prediction Task (Figure 10):** Identifies five distinct response patterns exhibited by models during probabilistic prediction.

- **Wisconsin Card Sorting Test (Figure 11):** Examines accuracy trends across six rule groups, revealing adaptation challenges when rules change.

- **Double Choice Iowa Gambling Task (Figures 12 & 13):** Analyzes short-term switching behaviors and long-term reward optimization strategies.

- **Probabilistic Reversal Learning Task (Figure 14):** Evaluates belief updating patterns following reward probability reversals.

- **Meta Bandit Task (Figures 15 & 16):** Investigates session-wise performance, highlighting difficulties in recognizing global reward patterns.

- **Prompting Strategy Analysis (Figure 17):** Explores the impact of different prompting strategies (free output, direct generation, and CoT) on task performance.

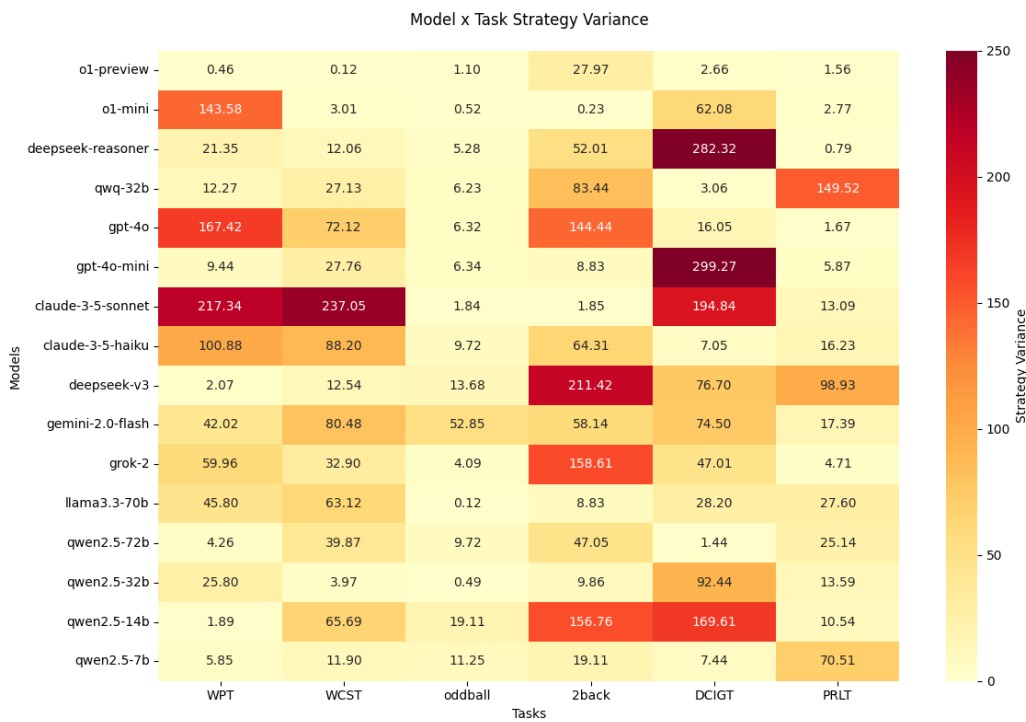

*Figure 9.* Model x Task Strategy Variance

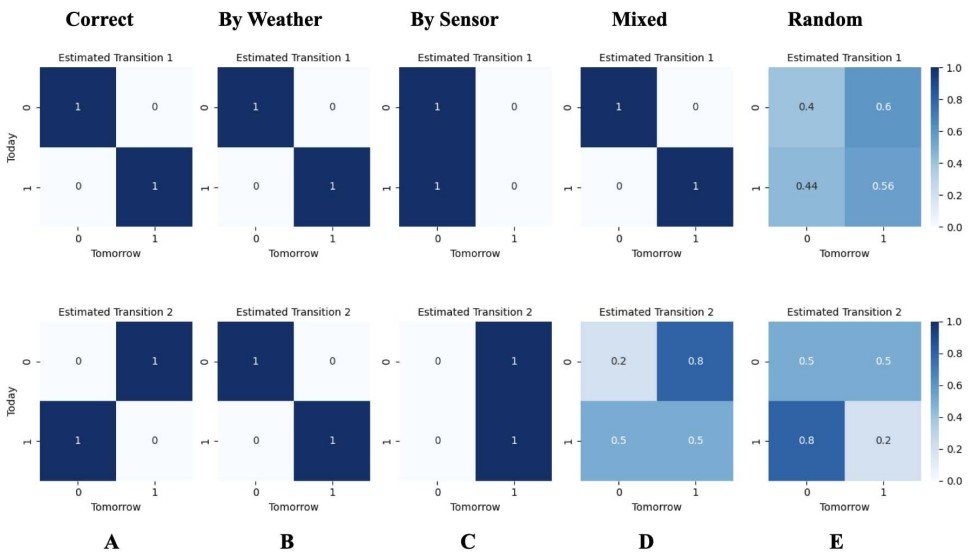

*Figure 10.* Five patterns of model responses

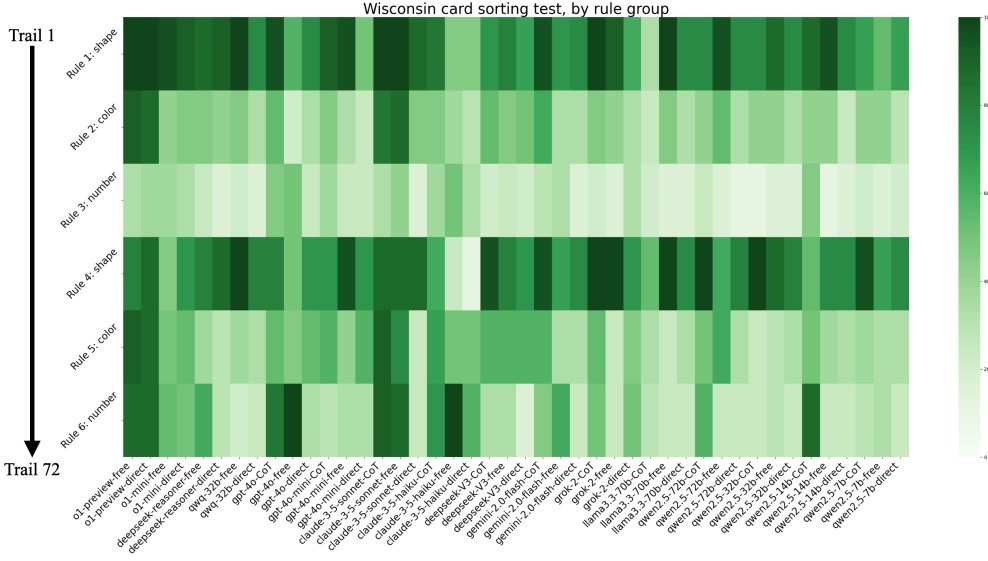

*Figure 11.* Wisconsin Card Sorting Test, accuracy by rule groups (6 blocks * 12 trials) over 72 trials

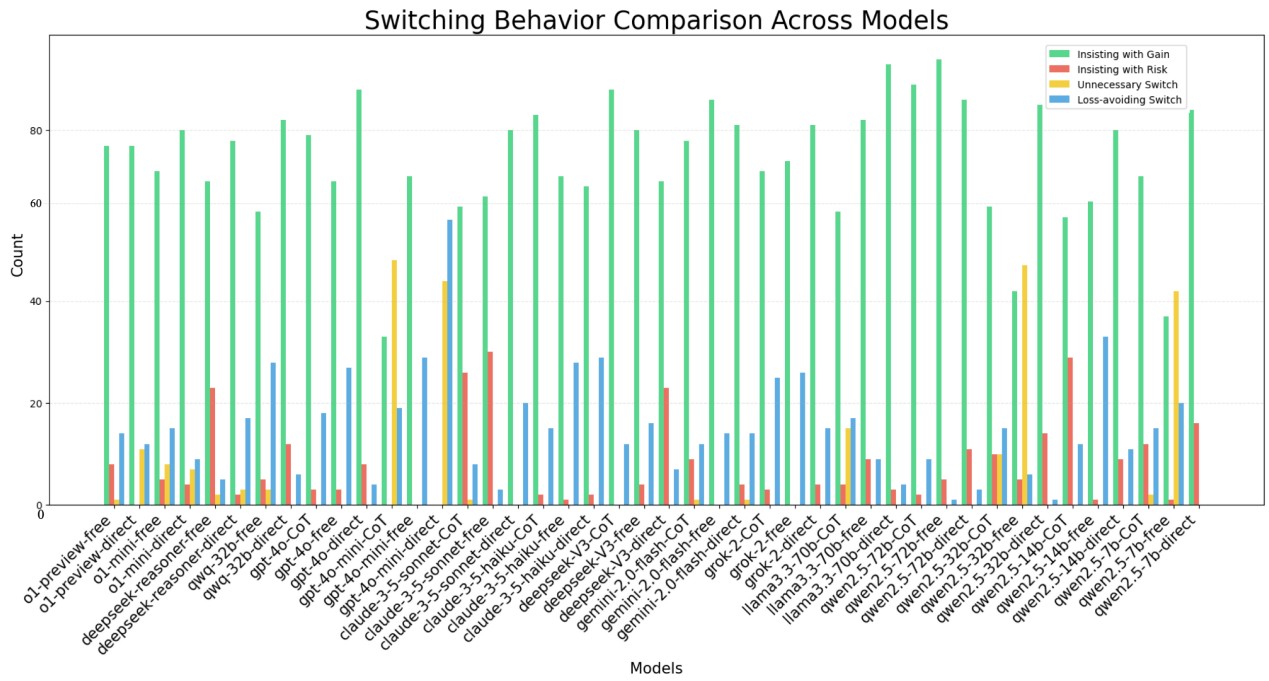

*Figure 12.* Short term: switching behaviors across models in the Double Choice Iowa Gambling Task

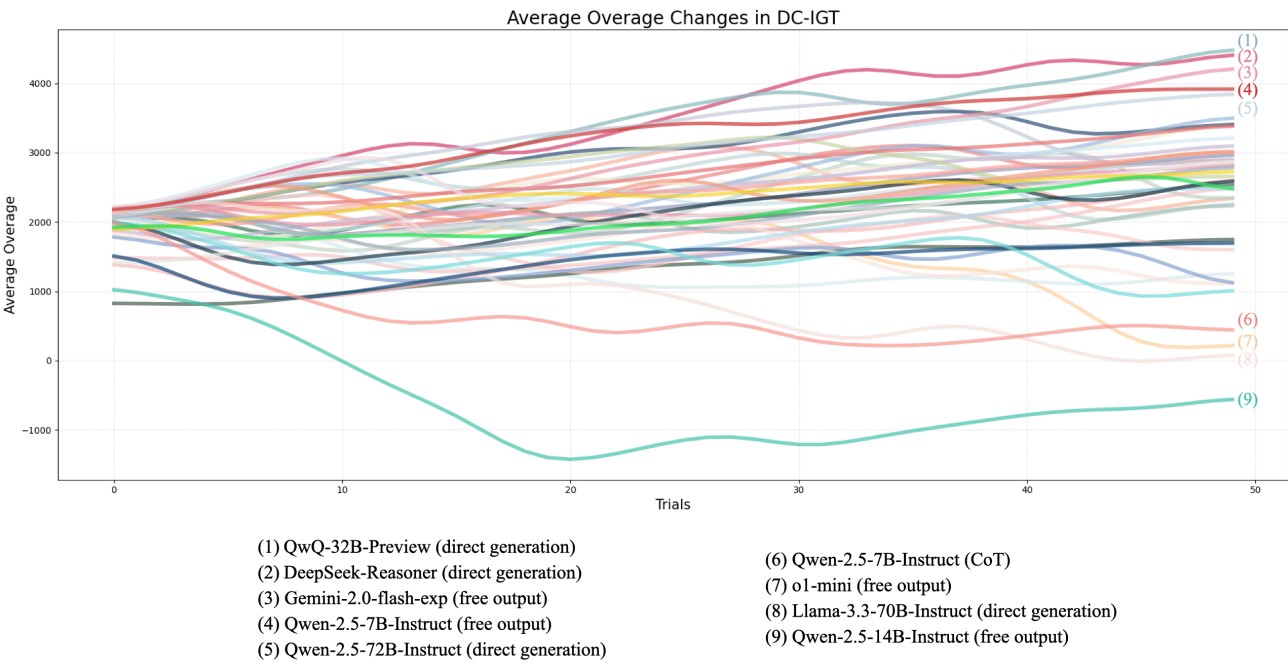

(1) QwQ-32B-Preview (direct generation)
(2) DeepSeek-Reasoner (direct generation)
(3) Gemini-2.0-flash-exp (free output)
(4) Qwen-2.5-7B-Instruct (free output)
(5) Qwen-2.5-72B-Instruct (direct generation)

(6) Qwen-2.5-7B-Instruct (CoT)
(7) o1-mini (free output)
(8) Llama-3.3-70B-Instruct (direct generation)
(9) Qwen-2.5-14B-Instruct (free output)

*Figure 13.* Long term: Average overage changes across models in the Double Choice Iowa Gambling Task

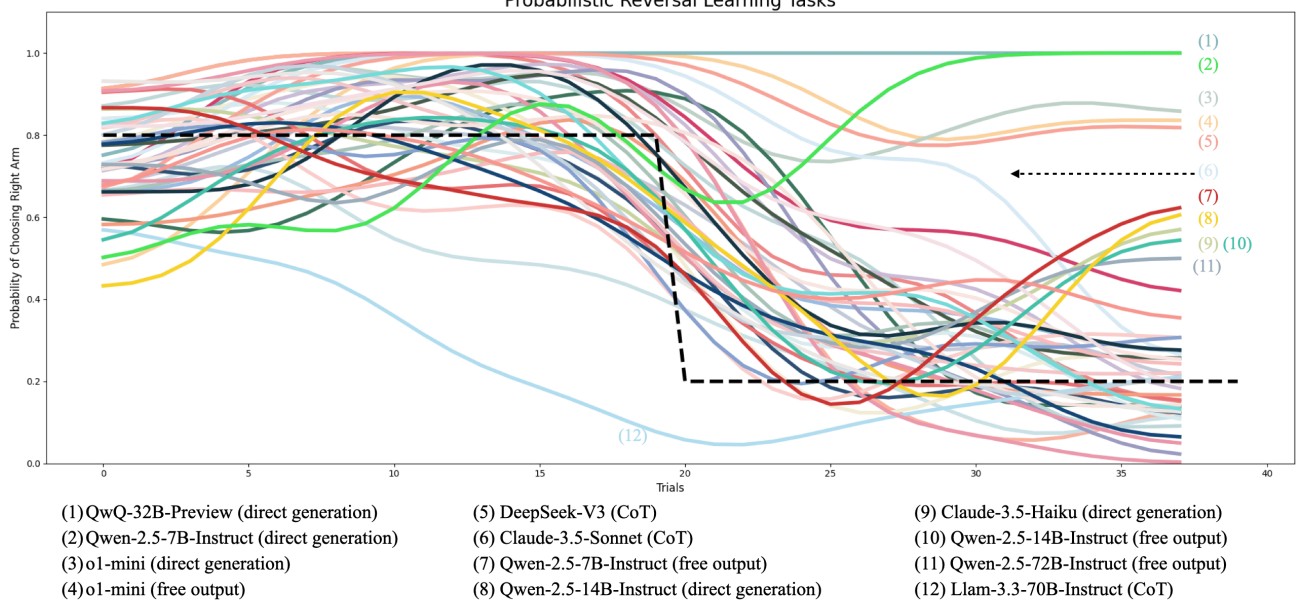

(1) QwQ-32B-Preview (direct generation)
(2) Qwen-2.5-7B-Instruct (direct generation)
(3) o1-mini (direct generation)
(4) o1-mini (free output)

(5) DeepSeek-V3 (CoT)
(6) Claude-3.5-Sonnet (CoT)
(7) Qwen-2.5-7B-Instruct (free output)
(8) Qwen-2.5-14B-Instruct (direct generation)

(9) Claude-3.5-Haiku (direct generation)
(10) Qwen-2.5-14B-Instruct (free output)
(11) Qwen-2.5-72B-Instruct (free output)
(12) Llam-3.3-70B-Instruct (CoT)

*Figure 14.* Probabilistic Reversal Learning Task

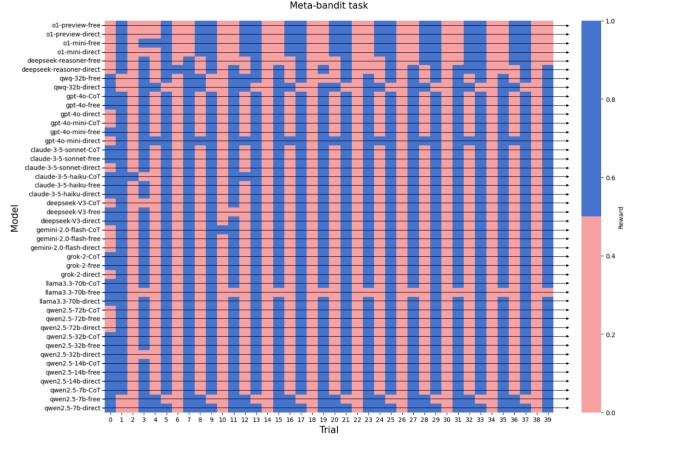

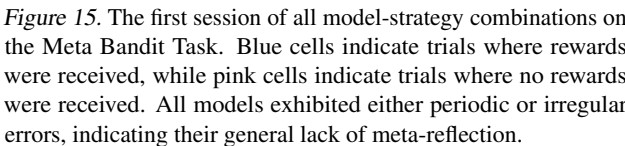

*Figure 15.* The first session of all model-strategy combinations on the Meta Bandit Task. Blue cells indicate trials where rewards were received, while pink cells indicate trials where no rewards were received. All models exhibited either periodic or irregular errors, indicating their general lack of meta-reflection.

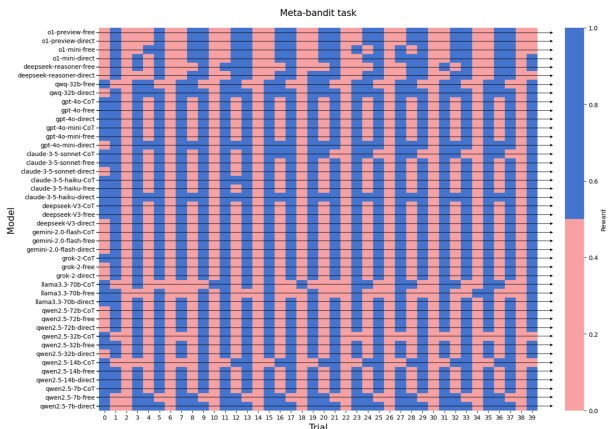

*Figure 16.* The second session of all model-strategy combinations on the Meta Bandit Task

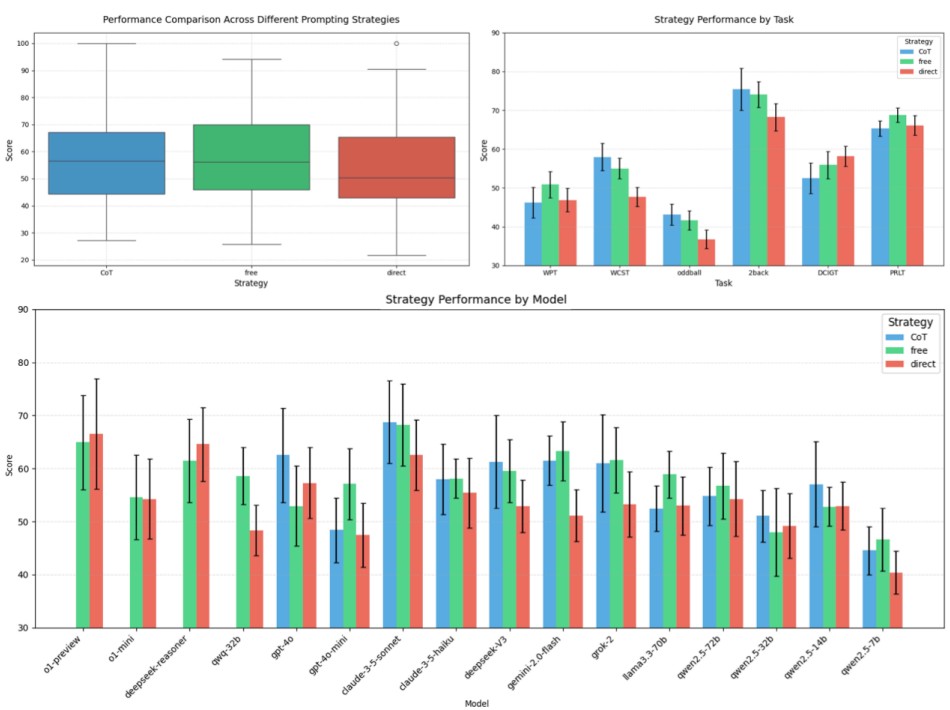

*Figure 17.* Performance analysis across different prompting strategies (overall, by task, and by model)

# D. Automated Evaluation of Oddball Test

## D.1. Rule-based Human Scoring

For the AAAA-B-AAA oddball prompt stimuli, the model's level of surprisals was annotated based on the following rule:

- 0: forced explanation or neglect B

- 1: simple enumeration of A and B

- 2: pointing out that A and B are different

- 3: clearly stating B is nonsensical in this sequence

We validated our automated evaluation method using 1,950 responses from 13 models, including o1-preview, o1-mini, GPT-4, GPT-4o, GPT-4o-mini, Claude-3.5-Sonnet, Gemini-1.5-Pro, Llama-3.1-405B/70B/8B-Instruct, and Qwen-2.5-72B/32B/14B-Instruct. Five of these models (GPT-4, Gemini-1.5-Pro, Llama-3.1-405B/70B/8B-Instruct) were not included in our main evaluation but were used to verify that our embedding-based method generalizes to other models. This approach ensures that our evaluation methodology applies to diverse models. These responses were scored by one Master's student in cognitive science.

### D.1.1. ANNOTATION EXAMPLE

**Oddball prompt**    The Great Wall of China is an ancient structure. It stretches over 13,000 miles. The wall was built for defense purposes. Many tourists visit it each year. ***Bananas are rich in potassium.*** Parts of the wall date back to the 7th century BCE. Some sections are well-preserved. The wall is visible from space.

**0: Forced explanation**    Wow, 13,000 miles is long! *Ancient defense against potassium-rich invaders?* Not sure about being visible from space, that's a myth. It's still amazing though!

**1: Simple enumeration**    The Great Wall of China is truly ancient and impressive. Over 13,000 miles is an astonishing length! Defense was a primary reason for its construction. It's clearly a major tourist attraction. *Interesting note about bananas being rich in potassium.* Some sections dating back to the 7th century BCE adds to its historical significance. It's good to know some parts are well-preserved. There's debate about its visibility from space, but it remains a popular claim.

**2: Pointing out differences**    The Great Wall of China is indeed a remarkable ancient structure. Stretching over 13,000 miles showcases its immense scale. It was primarily built for defense, highlighting its historical significance. Its popularity among tourists reflects its cultural importance. *Interesting fact about bananas being rich in potassium, though unrelated to the Great Wall.* Parts dating back to the 7th century BCE emphasize its long history. Well-preserved sections allow visitors to appreciate its original construction. The idea that the wall is visible from space is a common misconception.

**3: Clearly stating nonsense**    Interesting fact about the wall's age... *wait, what's with the bananas? That seems out of place.* Anyway, 7th century BCE is impressive. I'm not sure if it's entirely visible from space, though - I've heard that's a myth.

## D.2. Automated Evaluation Method

Since that, the surprises are reflected in the model's text responses. Intuitively, we can first craft a standard sentence expressing the highest level of surprises. Then higher similarity between the model's response and this standard surprise sentence can be defined as the level of surprise.

Specifically, by splitting the model's response into single sentences, the cosine similarity between the standard sentence and the model's responses is calculated. The highest cosine similarity is adopted as the score of this response.

We analyze the Pearson correlation between rule-based human annotation and our rule-free automated method to validate its efficiency. On single data points, the result indicates a good correlation between the two methods as shown in Figure 18.

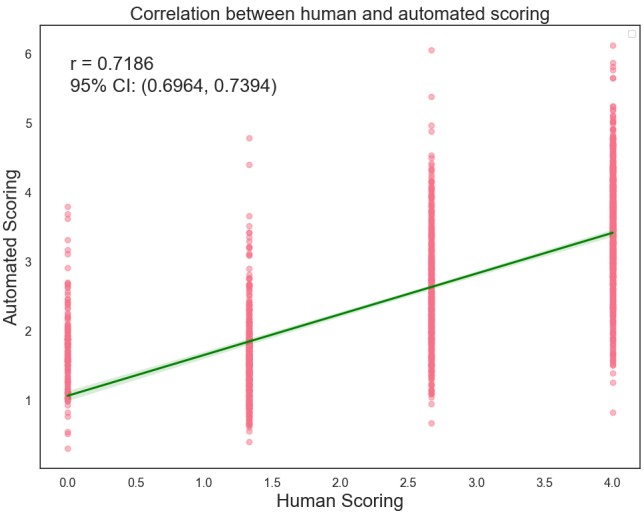

*Figure 18.* Correlation between human and automated scoring, single data point

Given that manual scores are ordinal categorical variables (0, 1, 2, and 3) while automated scores are continuous, we conducted a correlation analysis on aggregated data points to address this measurement discrepancy. As shown in Figure 19, the correlation strength increases with the level of aggregation. To balance the measurement continuity and statistical power, we drew our conclusion by aggregating 10 responses into one data point, resulting in 195 scores. It reveals a strong correlation between automated and manual scoring (r=0.87, p=5.83e-60), supporting the validity of this automated method.

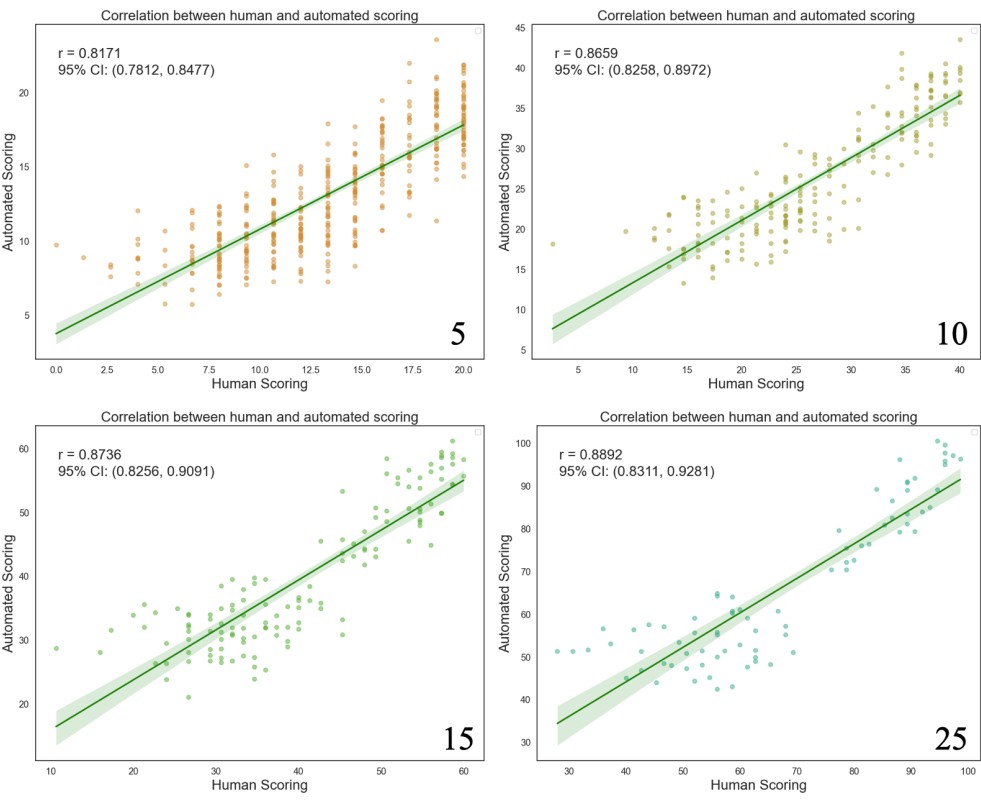

*Figure 19.* Correlation between human and automated scoring, aggregate data point

