# OpenReview forum: "Reflection-Bench: Evaluating Epistemic Agency in Large Language Models"
_ICML.cc/2025/Conference — ICML 2025 poster_

### Official Review · Reviewer_Xeen · 2025-03-12

**Overall Recommendation:** 2

**Summary:**

This paper proposes a cognitive-inspired benchmark, reflection-bench, to evaluate agency in LLMs. By decomposing the necessary cognitive procedures an agent would be required, the paper lists out seven important cognitive functions, including prediction, memory, belief updating, meta-reflection, and so on. The study selects a representative cognitive task from fields of psychology and cognitive science for each of the domains, and tests on multiple LLMs as well as several prompting strategies, such as CoT. The benchmark reveals models with larger sizes tend to perform much better than smaller sizes.

**Claims And Evidence:**

The paper has several claims regarding the benchmarks.

First, the paper claims the benchmark as a benchmark of agency. The evidence is that they decompose what an agent needs to do cognitively to interact with the environment, and test the cognitive functions with relevant but separate tasks. However, this kind of evaluation is hard to be called 'agency', since there is no agent in the benchmark at all. Though people have not reached complete consensus about the definition of agent, an agent typically should be able to use tools, learn and interact with the environment, goal-directed as well as plan for the goals, and of course, those dimensions already proposed by the paper. However, when evaluating an agent, we should place them in a real-world like environment to measure their capacity integratively, not separately. It's hard to say if a model performs all that well in the current agency benchmark can be called a good agent, since engaging in these tasks separately does not require agency at all. Therefore, agency, though can be composed of many cognitive properties, it is an integrative concept. Evaluating agency with separate cognitive domains does not make sense. In this sense, I can not agree that this benchmark is really evaluating agency.

Second, going specifically, the benchmark picks up seven representative cognitive tasks, corresponding to seven domains. However, when evaluating specific behaviors, only raw performance and some naive behavioral patterns are analyzed. The authors should go into more detail about the model behavior by applying computational models to describe and interpret their behaviors. In cognitive science and psychology, there are many computational models proposed. Using these models to fit parameters and even select a proper model could provide a deeper insight into LLM's behaviors.

Lastly, for each task, I do not know how many repeated experiments were conducted and how the configurations of LLMs, such as temperatures, top K, top P, are set up. Comparing that performance also does not reveal any statistical analysis to show meaningful differences. This also pins out the importance of going beyond purely behavioral performance, since some tasks like the probabilistic reversal learning task contain noises. Without a detailed look at behavioral patterns and statistical analysis, it's hard to say these models are really different in performance.

**Essential References Not Discussed:**

The paper has listed and discussed key literatures in the intersection of cognitive science and AI, especially important milestones and benchmarks in these fields.

However, in some detailed tasks, the citations of the reference may not be proper enough. For example, when citing references about the tasks, the authors should be aware of the origin of the task, not just cite a relevant paper. I have not gone through all the citations, but references below are definitely not accurate enough:

For example, in the Iowa Gambling Task, they should cite this:
Bechara, A., Damasio, A. R., Damasio, H., & Anderson, S. W. (2013). Insensitivity to future consequences following damage to human prefrontal cortex. In Personality and Personality Disorders (pp. 287-295). Routledge. (or 1994 the earlist version)
instead of this:
Buelow, M. T. and Suhr, J. A. Construct validity of the iowa gambling task. Neuropsychology review, 19:102–114, 2009. URL https://doi.org/10.1007/s11065-009-9083-4.

for the probabilistic reversal learning task, they should cite this at least (or earlier work):
Cools, R., Clark, L., Owen, A. M., & Robbins, T. W. (2002). Defining the neural mechanisms of probabilistic reversal learning using event-related functional magnetic resonance imaging. Journal of Neuroscience, 22(11), 4563-4567.

Since this benchmark is cognitive-inspired, necessary and accurate background about cognitive science and psychology would help the benchmark be established more properly.

**Experimental Designs Or Analyses:**

The tasks are mainly from the literatures in cognitive science and psychology, which are representative and mature. As I mention above, to better evaluate agency, the paper could be improved by testing on an integrative task, like a game (Allen et.al., 2024), to measure LLM's behaviors and cognitive functions integratively.

Some other approaches to deepen the design and analysis can be fine-tuning models (like Centuar, Binz et.al., 2024) on these tasks and testing them on alternative tasks, which also correspond to these concepts. This not only helps build up the understanding of how fine-tuning models can be used to improve agency, but also helps to evaluate how generalizable these selected tasks are.

Or the authors could go deeper into computational modeling analysis of behaviors, as well as look into how the model's internal representations support their specific behavioral patterns and strategies (with interpretability tools like SAE).

References:
Allen, K., Brändle, F., Botvinick, M., Fan, J. E., Gershman, S. J., Gopnik, A., ... & Schulz, E. (2024). Using games to understand the mind. Nature human behaviour, 8(6), 1035-1043.
Binz, M., Akata, E., Bethge, M., Brändle, F., Callaway, F., Coda-Forno, J., ... & Schulz, E. (2024). Centaur: a foundation model of human cognition. arXiv preprint arXiv:2410.20268.
Demircan, C., Saanum, T., Jagadish, A. K., Binz, M., & Schulz, E. (2024). Sparse autoencoders reveal temporal difference learning in large language models. arXiv preprint arXiv:2410.01280.

**Methods And Evaluation Criteria:**

Since it's a primarily benchmark paper, the evaluation and methods have been extensively discussed above.

**Other Comments Or Suggestions:**

I have no other comments and suggestions.

**Other Strengths And Weaknesses:**

Strengths: the paper examines extensive existing models, which are comprehensive and robust. The writing is clear and the visualization is neat and easy to understand.

**Questions For Authors:**

I current do not have any questions and will propose in the engagement of the rebuttal phase.

**Relation To Broader Scientific Literature:**

This benchmark proposes seven aspects of agency in cognitive domains, which are used to test extensive existing LLMs. They can somehow improve our understanding of these models' performance in these tasks. However, due to its lack of integration in these aspects in the task, as well as limited depth about analyzing those behaviors in a computational and neural (representational) aspect, the impact of the work is limited.

**Theoretical Claims:**

The paper does not involve theoretical proof, except the concept decomposition about agency. Though these cognitive dimensions make sense, the authors overlook that these dimensions should be integrated, not separated.

---

> ### Author Rebuttal · Authors · 2025-04-01
>
> Thank you for your thorough review and specialist insights. We deeply value your expertise and have carefully considered each point of your feedback.
>
> # "Agency" and its evaluation
>
> We appreciate your insightful critique regarding our "agency" conceptual framework. Your concerns about the terminology overlap of agency between our work and conventional agent capabilities is astute. Your comment that "... hard to be called 'agency', since there is no agent at all..." reflects a conception of agency centered on agent capabilities such as tool use. We would like to clarify that our focus is on the base model's foundational intrinsic quality, making these operational capabilities possible. In your recommended work Demircan et al., 2024, researchers compare Llama models with "human agents" and "Q-learning agents," suggesting that certain type of agency can be investigated where "there is no agent". Indeed, such kind of agency is undefined by the community yet, although they have gradually realized that base model's certain intrinsic quality significantly determines its effectiveness when deployed as an agent. This conceptual ambiguity highlights the need for more precise terminology to describe our scope.
>
> After careful literature review, we will refine our terminology from the broader "**agency**" to the more precise "**epistemic agency**" to better reflect our focus, e.g., changing the title to "Reflection-Bench: Evaluating Epistemic Agency in Large Language Models." Epistemic agency refers to the capacity to actively form and revise beliefs about external environments—evaluating evidence, updating beliefs with new information, and reflecting on belief-forming processes [1, 2, 3]. This concept more accurately captures what we evaluate: the cognitive processes underlying belief formation (prediction, decision-making, error detection, memory retrieval, counterfactual thinking, and belief updating) and meta-reflection on these processes.
>
> As we emphasize throughout our paper (Lines 21, 46-48, 140-145, 149), we fully agree that the seven cognitive dimensions are not separated but closely interconnected. Our methodological choice to evaluate them separately allows us to identify specific limitations that might be obscured in fully integrated tasks. By separately evaluating these cognitive dimensions—just as standardized cognitive assessments do in psychological research—we could establish necessary conditions for epistemic agency while providing targeted diagnostic insights that can guide focused improvements. Your comment that performance on separate tasks "does not require agency at all" might reflect a misunderstanding of our approach. While excellence in separated tasks doesn't constitute complete epistemic agency, deficits in any functions necessarily constrain a model's overall epistemic agency.
>
> We certainly recognize the value that integrative evaluations in game-like environments can provide, as demonstrated in recent work (Allen et al., 2024). We will discuss game-like evaluations as promising directions for future work.
>
> # Deeper analysis and experiments
>
> Considering our primary focus is establishing a standardized benchmark, we conducted 1 million random simulations across applicable tasks, establishing chance-level thresholds at the 95th percentile (Table 9). This provides a statistically sound metric for determining whether models merely produce plausible-looking outputs. We will incorporate it alongside our existing metrics. Thank you for helping enhance the scientific rigor of our work.
>
> Following your recommendation, we've added evaluation results for both Centaur and its base model on easy and hard settings. But Centaur shows no improvement, despite using specific prompt format as suggested. This unexpected result suggests Reflection-Bench reduces contamination. You can check the results at this anonymous link:  https://anonymous.4open.science/r/ICML_Rebuttal-773F/For%20Reviewer%20Xeen.md
>
> Your suggestion to apply computational models and SAE analysis is thoughtful. While such approaches provide additional insights, implementing these analyses expands our scope beyond establishing a comprehensive benchmark. For our task adaptations, new computational models would be needed. And behavioral failures present methodological challenges for internal representations analysis We anticipate that our benchmark will promote more such works in future.
>
> ### Other concerns
>
> We will enhance experimental clarity in our revised manuscript Section 4.1 and update the references to include original works accordingly.
>
> Thank you for your thorough and constructive review. We are grateful for your specialist comments, which strengthen our work regarding terminology refinement, statistical validation, and future research directions.
>
> [1] The Routledge Handbook of Philosophy of Agency
>
> [2] Knowledge, Dexterity, and Attention: A Theory of Epistemic Agency
>
> [3] Belief, Agency, and Knowledge Essays on Epistemic Normativity

---

> > ### Comment · Reviewer_Xeen · 2025-04-03
> >
> > Dear Authors,
> >
> > Thank you very much for your additional work. I appreciate the inclusion of Centaur as a comparable model and will raise my score to 2 in recognition of your responsiveness.
> >
> > However, I still cannot offer a more positive evaluation of the overall contribution. The central concern remains: the benchmark tasks, while inspired by cognitive constructs, do not constitute "agency"—even under the refined term "epistemic agency"—because they do not require the integration of multiple cognitive functions in a context where such integration is necessary. Without an agent interacting with an environment or pursuing goals, it is difficult to interpret these tasks as measuring any form of agency, epistemic or otherwise. The decomposition of agency into component tasks is useful, but separating them entirely limits the ecological validity of the evaluation.
> >
> > In addition, I find the claim that deeper analysis lies outside the scope of a benchmark to be somewhat inconsistent with the paper’s stated contributions. The benchmark is motivated by cognitive science, and the tasks are drawn from this literature. Therefore, it is reasonable to expect more cognitively meaningful analysis—such as computational modeling, learning curve characterization, or internal representation analysis—to support this positioning. Without such depth, the benchmark risks being another behavior-only evaluation, lacking the interpretive value that would make it stand out from existing work (e.g., Binz & Schulz, 2023).
> >
> > In short, the work remains limited both in **breadth** (its definition and implementation of agency) and **depth** (the insights it provides into model behavior). I appreciate the effort made to address these concerns, but they are, in my view, only partially resolved.
> >
> > Thank you again for the thoughtful engagement.

---

> > > ### Author Response · Authors · 2025-04-09
> > >
> > > Dear Reviewer,
> > >
> > > Thank you for your continued engagement and for recognizing our efforts. We appreciate your perspective while respectfully maintaining that our work makes meaningful contributions to the field.
> > >
> > > Our primary contribution lies in identifying and systematically decomposing epistemic agency - a previously ill-defined quality that fundamentally affects how base models perform when deployed as agents. This framework provides theoretical foundations for future research. As LLM-based agents emerge as the next frontier in AI research and applications, "epistemic agency" has broader implications for the trustworthiness of LLM-based agents - only when a model possesses robust mechanisms for belief formation and revision can it be considered accountable for its actions.
> > >
> > > Our empirical research evaluates current models' cognitive capabilities necessary for epistemic agency, offering fine-grained diagnostic insights for future development. Our current behavioral findings establish baselines while highlighting directions for model improvement. The observed poor performance of current models on several tasks, with some performing below chance level, inherently limits the utility of computational modeling and representation analyses at this stage. This underscores the timeliness of our behavioral assessment framework as a necessary first step, laying the groundwork for more sophisticated analyses once models demonstrate more robust capabilities.
> > >
> > > We look forward to this work developing as a research line that progressively incorporates more sophisticated evaluation methods and analyses, including the game-like evaluation, computational modeling, learning curve characteristics, and internal representation analyses you suggested.
> > >
> > > We sincerely appreciate your thoughtful feedback.

---

### Official Review · Reviewer_5qGQ · 2025-03-14

**Overall Recommendation:** 3

**Summary:**

The authors propose Reflection-Bench as a contamination-free benchmark consisting of seven parameterized cognitive tests inspired by cognitive psychology paradigms. The experimental evaluation spans 16 prominent LLMs and three prompting strategies: direct generation, free output, and Chain-of-Thought (CoT). Results identify a three-tier performance hierarchy among models, highlighting significant limitations particularly in meta-reflection. The paper concludes with implications for future research, notably enhancing meta-cognition, developing adaptive cognitive strategies, and encouraging coordinated cognitive capabilities within LLMs.

**Claims And Evidence:**

Yes

**Essential References Not Discussed:**

N/A

**Experimental Designs Or Analyses:**

The current tasks, while well-designed, remain somewhat abstracted from realistic, naturalistic contexts, potentially limiting their predictive power regarding real-world agent performance.

**Methods And Evaluation Criteria:**

- The authors only evaluated entry-level difficulty tasks. Evaluating different difficulty levels or scalability (e.g., medium and high difficulty) would strengthen the generalizability and robustness of results.
- While focused intentionally on intrinsic agency, the paper does not investigate how agency manifests within integrated, real-world agent workflows, limiting practical generalizability.

**Other Comments Or Suggestions:**

N/A

**Other Strengths And Weaknesses:**

**Strengths:**
1. **Comprehensive Framework:** Reflection-Bench is systematically designed, covering a robust set of cognitive dimensions rooted in cognitive psychology literature, and the tests are thoughtfully adapted to suit LLM evaluation contexts.

2. **Novelty and Significance:** The paper addresses the understudied aspect of intrinsic agency in LLMs, providing a comprehensive cognitive-level evaluation beyond traditional application-specific benchmarks.

3. **Clear Methodology:** The authors provide a detailed description of cognitive tests and their adaptation methods, ensuring clarity and reproducibility.

**Weaknesses:**

See Above

**Questions For Authors:**

N/A

**Relation To Broader Scientific Literature:**

N/A

**Theoretical Claims:**

Yes

---

> ### Author Rebuttal · Authors · 2025-04-01
>
> Thank you for your thorough review and recognition of our benchmark's strengths. We have addressed your key concerns as follows:
>
> # Evaluation across different difficulty levels
>
> We fully agree with your suggestion about evaluating tasks at varying difficulty levels. In response, we have expanded our evaluation to include more challenging parameter configurations. We tested 18 models (including the 16 original models plus Centaur, a model fine-tuned with human performances on various cognitive tests [1], and its base model Llama-3.1-70B-Instruct) on harder configurations using direct generation. The results, presented in a new table (Table 9), show the expected performance decreases on harder configurations and suggest that Reflection-Bench is both minimizing contamination and far from saturated.
>
> We also conducted 1 million random simulations across five applicable tasks, establishing robust chance-level thresholds at the 95th percentile of random performance distributions (Table 9). This provides a statistically sound metric for determining whether models merely producing plausible-looking outputs. Performance exceeding these thresholds indicates that models have developed meaningful understanding of the underlying task parameters. In our revised manuscript, we will incorporate these statistical benchmarks alongside our existing metrics. Thank you for helping enhance the scientific rigor of our evaluation framework.
> You can check Table 9 at this anonymous link: https://anonymous.4open.science/r/ICML_Rebuttal-773F/For%20Reviewer%205qGQ.md
>
> # Regarding real-world generalizability
>
> ## Definition Clarification
>
> Thank you for pointing out the concern about real-world applicability. While we would like to clarify that our research intentionally focuses on assessing models' intrinsic capabilities at the cognitive level, independent of specific external tools or applications. To further clarify our scope, we have refined our terminology "**agency**" to "**epistemic agency**," which more precisely captures our focus on cognitive capabilities that enable belief formation, revision, and reflection in dynamic environments -- making operational capabilities such as planning possible and reliable [2, 3, 4]. Undoubtedly, without robust intrinsic epistemic agency, even the most sophisticated tool integration or workflow design will be limited by the model's core cognitive constraints and, therefore, not reliable. The research community has increasingly recognized that LLM-based agent performance critically hinges on some intrinsic quality of the base model, yet there remains considerable ambiguity about the precise nature of this quality. Our benchmark aims to evaluate it as "epistemic agency" that determines whether models can serve as reliable cores for AI agents in any real-world context. We will revise our manuscript accordingly and update the title to "Reflection-Bench: Evaluating Epistemic Agency in Large Language Models." We believe that by establishing "epistemic agency" as a well-defined, measurable characteristic of language models, our work provides the missing framework to systematically identify and evaluate this crucial but previously nebulous quality.
>
> ## Ecological validity
>
> As we emphasized in line 76-80:
>
> "Cognitive tests create controlled environments where subjects must learn and reason about unknown parameters through interaction, offering standardized, quantified, and objective assessment tools that mirror real-world functioning. ",
>
> cognitive tests are specifically designed to extract and evaluate the abstract cognitive features underlying everyday real-world functioning. This property, known as "ecological validity" in cognitive assessment, allows controlled tasks to provide meaningful insights into fundamental capabilities that support real-world performance. Similarly, our adapted tests extract core cognitive dimensions essential for epistemic agency in any environment. We will acknowledge in our Limitations section that the ecological validity of Reflection-Bench for LLMs specifically requires further validation, and identify this as an important direction for future work. We will also discuss how future iterations could incorporate more naturalistic contexts to complement our controlled assessments.
>
> We are grateful for your thoughtful feedback, which significantly strengthens both the conceptual clarity and methodological rigor of our research, enhancing Reflection-Bench's contributions to the field by providing a comprehensive framework for evaluating the foundational capabilities that ultimately determine an LLM's effectiveness as a reliable agent core.
>
> [1] Centaur: a foundation model of human cognition
>
> [2] The Routledge Handbook of Philosophy of Agency
>
> [3] Knowledge, Dexterity, and Attention: A Theory of Epistemic Agency
>
> [4] Belief, Agency, and Knowledge Essays on Epistemic Normativity

---

> > ### Comment · Reviewer_5qGQ · 2025-04-07
> >
> > Thanks for the clarification. I have raised my score.

---

### Official Review · Reviewer_EAfK · 2025-03-15

**Overall Recommendation:** 4

**Summary:**

This paper proposes a benchmark to evaluate agency in large language models. The authors define agency along seven dimensions, namely prediction, decision-making, perception, memory, counterfactual thinking, belief updating, and meta-reflection. For each of these, the authors adapt a task from cognitive psychology for LLMs. The benefit of these tasks is that they are parametrised and 'contamination-free' (in the sense that if a model has memorised a task you can change the parameters, and also that the tasks for a particular parameter configuration are unlikely to occur in training data). The authors evaluate many LLMs in three classes (normal LLMs, reasoning LLMs, and Qwen models), with three different prompting strategies (direct generation, free generation, and CoT), and repeat each task at least twice. They find that their benchmark has a clear discriminating factor, and models exhibit some agency according to their measure, but fail on some tasks. Most notably, no model is able to perform the meta-reflection task which requires determining a repeating pattern in a sequence and adapting predictively based on it.

## Update after rebuttal
My main points are addressed, and I am in favour of accepting this paper.

**Claims And Evidence:**

Claims are supported.

**Essential References Not Discussed:**

N.A.

**Experimental Designs Or Analyses:**

Everything seems sound/valid.

**Methods And Evaluation Criteria:**

The methods and evaluation criteria make sense. The evaluated dimensions are all important for agency, although they do not encompass every aspect that is commonly considered agentic (e.g. planning). The authors evaluate a comprehensive set of models, use different prompting techniques, and repeat experiments to handle stochasticity.

**Other Comments Or Suggestions:**

- Consider highlighting that the definition of agency is a well-studied and difficult subject without consensus, and the presented definition here is not generally accepted.
- l160-162 needs to be rewritten
- Would be useful to see a figure or table with an example for each task in 3.2
- The number reflecting the result is hard to interpret. Can you say what the interpretation is? Would it be useful to get a human score on this benchmark for comparison?
- Styling error: When referring to figures like "Figure K", the space is often missing (e.g. see Line 351r and 315l)

**Other Strengths And Weaknesses:**

**Strengths**
- The tasks are original, well-adapted for LLMs, and the results can discriminate between LLMs well
- For one task currently all LLMs fail (the meta-reflection task)
- The authors do a comprehensive evaluation (multiple models, trials, and prompting strategies)
- The authors do a comprehensive analysis of the results, and do interesting findings such what kind of strategies models employ for sequential belief updating tasks (namely, that models employ a win-stay-lose-switch strategy)

**Weaknesses**
- Presentation of results in figures. Figure 14 and 15 are difficult to parse. Consider also using informative captions. For Figure 16, you're connecting the dots with lines between models. But the points on the line between models don't indicate anything. For categorical X-labels don't use a line plot.
- The analysis is only done for one set of parameters; it would be interesting to have at least one more set of parameters to see how models get worse/better for different parameters in the tasks.

**Questions For Authors:**

- Appendix D.1; authors validate automatic evaluation with a human. I can't parse *"Five of them were out of the evaluated models in this paper for verifying the generality of our automated evaluation method based on text-embedding-3-large."* How many examples do you evaluate? And only of 5 models of 13? How were these selected?
- Figure 18; what does each panel refer to?
- How is CoT and direct generation done? I can't find the prompts in the appendix.
- Why is Qwen a separate category of models?
- Interesting analysis for the Wisconsin card sorting test; models not being able to change from the shape rule they determined ("shape sink"). could this be connected to a shape bias in humans (Landau et al. 1988). E.g., maybe models could change to different rule if it does not concern a shape rule, did you try this?

**Relation To Broader Scientific Literature:**

Many recent works evaluate agency in LLMs, but the authors distinguish themselves by defining agency along 7 dimensions and taking tasks from cognitive psychology and adapting them to LLMs.

**Theoretical Claims:**

N>A.

---

> ### Author Rebuttal · Authors · 2025-04-01
>
> Thank you for your thoughtful review and helpful suggestions. We've addressed your feedback as follows:
>
> # Complementary experiments
>
> Following your suggestion, we evaluated 18 models (original models plus two additional models: Centaur,  fine-tuned with human cognitive tests performances [1], and its base model Llama-3.1-70B-Instruct) on a difficult parameter set using direct generation. The results (Table 9) show expected score decrease, suggesting that Reflection-Bench is both leakage-resistant and far from saturated.
>
> As for the score interpretation, we conducted one million random simulations for five applicable tasks (excluding the non-parameterized oddball task and qualitative meta-bandit task) to establish chance-level thresholds (95th percentile). Scores above these thresholds (Table 9) indicate statistically significant task performance, with higher scores reflecting a more precise inference of task parameters.
>
> # Definition of agency
>
> Thank you for your feedback on our conceptual precision. We will add a paragraph recognizing the conceptual complexity of "agency". To further address the ambiguity between "agency" and common agent capabilities like planning and tool usage, we will revise the terminology to "**epistemic agency**", a concept that more accurately captures our research scope. It refers to one's intrinsic cognitive foundation for constructing, modifying, and monitoring its beliefs about the external world [2, 3, 4], independent of specific external modules or tools. Additionally, epistemic agency has direct implications for AI trustworthiness -- only when a model possesses robust mechanisms for belief formation and revision can it be considered accountable for its actions.
>
> We believe this refinement strengthens the theoretical precision and potential contributions of this work. While the research community widely acknowledges that some intrinsic quality of base models significantly determines their effectiveness as agents, this characteristic remains inadequately characterized (sometimes simply labeled as "intelligence"). Our benchmark aims to characterize and measure this elusive foundation, .i.e. epistemic agency, which will contribute meaningfully to the field.
>
> # Presentation improvements
>
> We've enhanced the presentations by:
> - Redesigning Figures 14 & 15 with more informative captions
> - Replacing line plots in Figure 16 with bar charts
> - Adding a comprehensive figure in Section 3.2 with illustrations for all task (Figure Tasks)
> - Fixing spacing issues in figure references throughout the paper
> - Rewording lines 160-162 for clarity: "For LLMs, prediction capabilities are crucial for planning tasks, where models must reason about which policies will effectively transition an agent from its initial state to a desired goal state."
> - adding the prompts for two strategies in Appendix A
>
> # Other specific questions
>
> - Appendix D.1 clarification: We validated our automated evaluation method using 1,950 responses from 13 models. Five of these models (GPT-4, Gemini-1.5-Pro, Llama-3.1-405B/70B/8B) were not included in our main evaluation but were used to verify that our embedding-based method generalizes to other models. This approach ensures that our evaluation methodology is applicable to a diverse range of models.
> - Figure 18 panels: Each panel represents different levels of data aggregation (from 5 to 25 datapoints) used to analyze the correlation between human annotation and automatic evaluation.
> - Qwen categorization: We separated the Qwen-2.5 series (72B, 32B, 14B, 7B) as one category to show how performance scales with model size within a consistent model family.
> - Human baselines: Our adapted tasks differ from standard human cognitive tests, making direct comparisons with existing human scores potentially not appropriate. We will add a discussion in the limitations section about the human-LLM comparisons for establishing the ecological validity of Reflection-Bench.
> - Shape bias in WCST: This is an insightful observation. We conducted additional experiments with four Qwen-2.5 models where we modified both the rule blocks and card formats from shape-color-number to color-number-shape. We found that the "shape sink" effect persists for Qwen2.5-14B-Instruct (Figure WCST), while less evident for the other three models. Although beyond the scope of the current paper, further investigation of this phenomenon could provide valuable insights into how language models indirectly encode human cognition in other modalities.
>
> Thank you again for your valuable feedback, which has significantly improved our paper.
>
> You can view the updated table and figures at this anonymous link: https://anonymous.4open.science/r/ICML_Rebuttal-773F/For%20Reviewer%20EAfK.md
>
> [1] Centaur: a foundation model of human cognition
>
> [2] The Routledge Handbook of Philosophy of Agency
>
> [3] Knowledge, Dexterity, and Attention: A Theory of Epistemic Agency
>
> [4] Belief, Agency, and Knowledge Essays on Epistemic Normativity

---

### Official Review · Reviewer_sNx4 · 2025-03-17

**Overall Recommendation:** 3

**Summary:**

This paper presents Reflection-Bench, a benchmark designed to evaluate the intrinsic agency of LLMs from seven cognitive dimensions: prediction, decision-making, perception, memory, counterfactual thinking, belief updating, and meta-reflection. The authors use or adapt a cognitive psychology-inspired and parameterized test for each of the seven dimensions, and evaluate 16 LLMs across three model categories, using three prompting strategies. The performance distribution shows a clear three-tier hierarchical performance structure aligned with model scaling, demonstrating a basic level of agency. However, detailed behavioral analyses reveal significant weaknesses in LLMs’ capabilities, particularly in prediction, decision-making, and meta-reflection. The results suggest that future research should focus on improving meta-cognition, developing dynamic reasoning mechanisms, and enhancing coordination among cognitive capabilities.

**Claims And Evidence:**

* The authors claim that by using parameterized tests (parameters remain dynamic or unknown while the task might be seen during training), this benchmark is "contamination-free" --- I believe this is an over-statement. Although this may avoid verbatim memorization, having seen the task format with alternate numbers or solutions still counts as implicit contamination ("implicit contamination" in [1], footnote 2; "in-distribution contamination" in [2]). While I agree that parameterized tests may greatly reduce such contamination, "contamination-free" is over-claiming (in fact, in section 6, a weaker statement is used: "ensures minimization of potential data contamination").

* Another problematic claim (or general perspective) is that this benchmark is claimed to measure intrinsic "agency". A critical aspect of LLM agents, (autonomous) tool use, is completely missing.

Other claims are supported by clear and convincing evidence.


[1]. Generalization or Memorization: Data Contamination and Trustworthy Evaluation for Large Language Models

[2]. DICE: Detecting In-distribution Contamination in LLM’s Fine-tuning Phase for Math Reasoning

**Essential References Not Discussed:**

To my best knowledge, no essential reference is missing.

**Experimental Designs Or Analyses:**

The experimental designs and analyses are sound. The cognitive tests are originated from human cognition literature and the adaptations are reasonable for the purpose of testing the cognitive process of LLMs. Model-based automatic evaluation is validated with detailed correlation checks with human annotations. The model and prompt selection are also reasonable.

**Methods And Evaluation Criteria:**

The proposed methods and evaluation criteria mostly make sense for the problem or application at hand. However, I believe it is not adequate to fully evaluate "agency", as mentioned above.

**Other Comments Or Suggestions:**

* To provide a more compiling evidence of "contamination-free", specific testing protocols could be incorporated (e.g. [3]), and new cognitive tests/games (with newly-created rules) or tests after the model's cutoff time can be developed (e.g. similar to [4]), although I do acknowledge that this might be beyond the scope of the current work (i.e. I would be happy if the authors simply weaken this statement for this work). Therefore, I put it here as a suggestion for future versions.


[3]. Investigating Data Contamination in Modern Benchmarks for Large Language Models

[4]. LiveBench: A Challenging, Contamination-Free LLM Benchmark

**Other Strengths And Weaknesses:**

Strengths:

* This benchmark adapted several cognitive tests to evaluate LLM agency, which appears a novel contribution.

* The analyses (including the secondary analysis on the model performance patterns) are in-depth, highlighting clear trends in decision-making, memory, and meta-reflection capabilities.

* The results provide interesting insights into critical future directions.

Weaknesses:

* Overclaiming or inadequate evidence for "contamination" and "agency".

**Questions For Authors:**

1. For the MBT tests for meta-reflection, which the models generally struggled on, do you have some failure examples (showing model output) and detailed error analysis on the failure patterns?

**Relation To Broader Scientific Literature:**

This benchmark builds upon existing research in LLM evaluation, cognitive psychology, and AI agency. Prior work on LLM evaluation has largely focused on individual and task-specific benchmarks (e.g., reasoning, planning, and decision-making) rather than holistic agency, while Reflection-Bench frames these abilities within a structured agency evaluation framework.

**Theoretical Claims:**

Not applicable.

---

> ### Author Rebuttal · Authors · 2025-04-01
>
> Thank you for your time and thoughtful suggestions. We've addressed your concerns as follows:
>
> # Regarding "contamination-free" claims
>
> We agree that our claim of "contamination-free" was overstated. We will revise all such instances throughout the paper to use more precise language such as "reducing potential data contamination" or "minimizing data leakage."
>
> To address this concern further, we examined your recommended references and considered applying the TS-Guessing method [1], but found it incompatible with Reflection-Bench's format where models typically choose between given strucuralized options (Right/Left, Yes/No). We've also added new evaluation results of Centaur [2] (specifically fine-tuned on human cognitive test performances) and its base model Llama-3.1-70B. Centaur shows no performance improvement compared to its base model, providing empirical evidence that our parameterized design effectively minimizes data leakage concerns. Additionally, we will add a discussion in our Limitations section about developing novel test designs similar to LiveBench [3] as a promising direction for future work.
>
> # Clarifying the scope of "agency"
>
> We agree with the potential confusion between our notion of agency and agent capabilities such as planning and tool usage. Our focus is not on these operational capabilities but rather on the underlying processes that make such capabilities possible and reliable. While there is a consensus that base model's certain intrinsic quality significantly determines its effectiveness when deployed as an agent, this foundational characteristic remains poorly characterized in the research community (someone calls it "intelligence"). Our paper aims to identify this undefined quality and evaluate it systematically.
>
> Based on your feedback, we have refined our terminology to "**epistemic agency**," which more accurately captures our intended meaning. This philosophical concept refers to a model's intrinsic cognitive foundation for constructing, modifying, and monitoring its beliefs about the external world [4,5,6] - independent of specific external modules, tools, or applications. Moreover, epistemic agency has direct implications for the trustworthiness of AI systems. Only when a model possesses robust mechanisms for belief formation and revision can it be considered accountable for its actions.
>
> In the revised manuscript, we will clarify this distinction throughout, including changing the title to "Reflection-Bench: Evaluating *Epistemic Agency* in Large Language Models." We believe this refinement addresses your concerns. Thank you for helping us clarify this crucial aspect of our work, which strengthens both the conceptual precision and potential contributions of our research.
>
> # MBT failure example
>
> We had provided a representative example of MBT (meta-bandit task) failure in Figure 7, but did not explicitly point out in its caption. So we clarified the caption of Figure 7 to explicitly identify it as showing representative failure patterns from GPT-4o (Figure 7).
>
> You can check the revised Table 9 and Figure 7 at this anonymous link: https://anonymous.4open.science/r/ICML_Rebuttal-773F/For%20Reviewer%20sNx4.md
>
> # Code and data availability
>
> We confirm that the complete code and dataset for Reflection-Bench will be open-sourced on GitHub following the double-blind review process.
>
>
> We believe these revisions address your concerns while improving the paper's rigor and clarity. Thank you again for your valuable contributions to strengthening our work.
>
> [1] Investigating Data Contamination in Modern Benchmarks for Large Language Models
>
> [2] Centaur: a foundation model of human cognition
>
> [3] LiveBench: A Challenging, Contamination-Free LLM Benchmark
>
> [4] The Routledge Handbook of Philosophy of Agency
>
> [5] Knowledge, Dexterity, and Attention: A Theory of Epistemic Agency
>
> [6] Belief, Agency, and Knowledge Essays on Epistemic Normativity

---

> > ### Comment · Reviewer_sNx4 · 2025-04-03
> >
> > Thank you for your response on data contamination and the scope of "agency". With the current scope and contribution effectively reduced to "testing the intrinsic cognitive capabilities of LLMs", I find it quite borderline. Therefore, I am inclined to keep my score unchanged.

---

### Decision · Program_Chairs · 2025-05-01

**Decision:**

Accept (poster)

**Comment:**

The authors introduce Reflection-Bench for evaluating “agency” in LLMs. The evaluation is based on seven cognitive tests designed to avoid dataset contamination for fair evaluation. The work nicely outlines ontologically what is involved in the notion of agency though there is still a potential mismatch in the reader’s expectation and the authors’ intention. In particular, reviewers expect certain abilities/definitions including planning and tool use which define a more robust and real world evaluation while the authors claim this is misaligned with their work’s aims – epistemic. The reviewers believe that a revised submission which more accurately describes the assumptions of the work will lead to a more accurate read. Aside from framing, there is a concern about contamination – an issue that has come up with other cognitive tests that have undergone permutation for evaluation (e.g. Sally Anne variants). This likely also requires careful phrasing to ensure an accurate representation of the results.